# LOW-RANK INTERPRETABLE CELL-CELL HIDDEN INTERACTIONS FROM EMBEDDINGS

## ABSTRACT

Multicellular organisms rely on continuously changing cell–cell interactions that govern critical biological processes as cells modify their internal states and trajectories in space over time. Studying these interactions is critical to understand development, homeostasis, and disease progression. Live-cell imaging provides a unique opportunity to directly observe these dynamical events; however, current computational approaches often fail to model complex, time-varying events involving diverse populations and spatial contexts. Here, we present LICCHIE, a model designed to infer time-changing, feature-based cell-cell interactions, applicable across systems and conditions. Our approach represents each cell with a dynamic multi-feature vector, and interactions are modeled as spatially constrained, directed influences between cell pairs, evolving over time. We optimize the model using an iterative scheme balancing data fidelity, interactions smoothness, and low-rank sparse structure. We validated LICCHIE's ability to capture cellular interactions across populations in a controlled synthetic setting, and applied it to real-world 3D live-cell imaging of patient-derived tumor organoids to (1) identify components with interpretable structures that capture interaction type and directionality, and (2) suggest modulation strategies that may accelerate Natural Killer (NK) cells polarization and tumor cell death.

# 1 INTRODUCTION

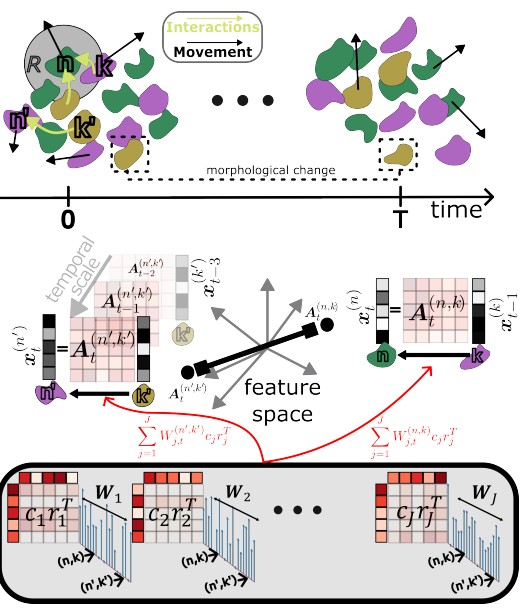

Figure 1: *LICCHIE learns cell-cell interactions from temporal trajectories.* **Top**, Cells from diverse populations can adapt their morphology, internal state, and location in response to cross-cell interactions and environmental effects. **Middle-left**, Interaction matrices, $\{\boldsymbol{A}_t^{(n,k)}\}$, represent the effect of cell $k$ on cell $n$ at time $t$, capture pairwise time-changing linear effects via cell features, constrained on feature-space distance for smooth, interpretable changes. **Middle-right**, time-varying vector $x_t^{(n)} \in \mathbb{R}^{m^{p(n)}}$, captures cell-specific attributes. The interaction matrices map features of nearby cells at $t-1$ to source feature at time $t$, providing temporal consistency. **Bottom**, Underlying matrices $\boldsymbol{A}_t^{(n,k)}$ emerge from a set of fixed rank-1 processes representing source $\{\boldsymbol{c}_j\}$ and target $\{\boldsymbol{r}_j\}$ effects, modulated via flexible weights to shape the overall transition matrix $\boldsymbol{A}_t^{(n,k)} = \sum_{j=1}^J W_{j,t}^{(n,k)} \boldsymbol{c}_j \boldsymbol{r}_j^T$, capturing complexity while maintaining parsimony.

Cell–cell interactions are central to the function and organization of multicellular systems (Su et al., 2024). Disruption or mis-regulation of communication is implicated in cancer, autoimmune disease, and developmental disorders (Armingol et al., 2021; Liu et al., 2024). Capturing how one cell's state and behavior influence another in real time is challenging since interactions span spatial and temporal scales, and depend on cell type and state.

**Studies based on static snapshots reveal a partial story.** Single-cell sequencing technologies have enabled systematic inference of putative ligand–receptor activity and co-variation patterns, offering a powerful view of communication at scale (Wilk et al., 2024; Heumos et al., 2023). However, such data are typically dissociated and time-agnostic, and methods built on them suffer from an inherent limitation: they recover statistical associations from static measurements rather than the dynamics of how cells influence each other in real time (Armingol et al., 2024; 2021; Wagner et al., 2016). As a result, they fail to reveal how cells change as a result of interactions and struggle to expose directionality, temporal delays, or state-dependent effects that are essential to mechanistic understanding (Weinreb et al., 2018).

**Live-cell imaging: a closer observational proxy for cellular dynamics.** Live-cell imaging provides a means to overcome many of the challenges in studying cell–cell interactions. By tracking individual cells over time, such data allow observation of cell states before, during and after interactions, and thus offer a window into the dynamic consequences of cellular communication (Cuny et al., 2022). Such longitudinal data provides ground truth for understanding how cells respond to given interactions and interventions, allowing for advanced therapeutic applications (Alieva et al., 2023). Hence, there is a need for novel computational models that can: (1) detect interactions, (2) quantify their strength, and (3) reveal how they promote changes in cell state.

**Our contributions.** We introduce **LICCHIE**—**L**ow-rank, **I**nterpretable **C**ell-**C**ell **H**idden **I**nteractions from **E**mbeddings—a data-driven method to infer time-varying, feature-based interaction rules from live-cell imaging. We represent each cell with a dynamic feature vector encompassing spatial and internal states. Direct interactions between source–target cell pairs are modeled as temporally-evolving linear transitions over features, active only within a spatial radius between cells and regularized to vary smoothly within the feature space. This formulation enforces temporal consistency across consecutive time-points while operating in a generalizable feature space. To balance expressivity with interpretability, LICCHIE decomposes pairwise interaction matrices into a few rank-1 components, with time- and cell-pair-varying weights capturing context, while keeping the components biologically readable. We jointly estimate the interaction matrices, shared components, and sparse weights in an iterative approach (Fig. 1). More precisely, we present:

- A modeling framework that detects direct, time- and cell-varying interactions at the feature level (LICCHIE).
- A low-rank decomposition into shared, interpretable source/target motifs with sparse, context-varying weights.
- Validation on synthetic data and 3D tumor–NK imaging, recovering time- and cell-varying interactions with interpretable structure.
- Identification of interactions that can influence NK polarization and tumor death using engineered observations, capturing unseen data points.

## 2 BACKGROUND

**Representation learning and static graphs.** Deep learning architectures, such as autoencoders have been used to discover interaction-relevant structures in single-cell omics but often yield embeddings that are hard to interpret mechanistically for interactions (Alessandri et al., 2021; Ternes et al., 2022). Graph neural networks (GNNs) encode cells as nodes and potential interactions as edges, yet typically presume a predefined (often symmetric) graph and still obscure direct, dynamic effects; attention models can highlight dependencies but do not directly encode interaction signals (Lazaros et al., 2024; Tang et al., 2023). Together, these tools are not tailored to direct, time-varying, feature-level effects required for dynamic inferences.

**Latent dynamical systems.** In neuroscience, switching linear dynamical systems capture abrupt regime changes (Linderman et al., 2017; Nassar et al., 2018), and decomposed LDS variants capture smoothly time-evolving interactions (Mudrik et al., 2024; Chen et al., 2024). However, these models do not directly address live-cell specifics: distance-modulated effects, transient visibility, strong state changes (e.g., death/polarization), division/proliferation, and motility that reshapes neighborhoods—motivating an imaging-first, interpretable model at per-cell feature resolution.

**Requirements for an imaging-based interpretable model.** We seek a framework that (i) models direct, feature-level influences between individual cells; (ii) captures non-stationarity over time and allows for other sources of interaction variability (e.g., distances, temporal resolutions), (iii) is spatially constrained and smooth in feature space for robustness; (iv) uncovers the core interpretable structures underlying the dynamics to support biological understanding of the interactions; and (v) does not rely on persistent cell identities, by operating directly in feature space. LICCHIE is designed around exactly these requirements (Sec. 4, Fig. 1).

## 3 PROBLEM FORMULATION

**Observations and notation.** We observe a system of $N$ interacting cells over $T$ frames, where each cell $n = 1 \ldots N$ is defined via a position vector (in 2D/3D space) $\{\boldsymbol{\psi}_t^{(n)}\}_{t=1}^T \in \mathbb{R}^s$, and a multi-feature, time-varying vector $x_t^{(n)} \in \mathbb{R}^{m^{p(n)}}$ capturing cell-specific features, defined by the biological system observed. In what follows, we relate to the multi-dimensional feature vector of each cell at time $t$ as the "cell-state". A system can consist of one or more populations, such that $p(n)$ indexes the population type of cell $n$; therefore, $m^{p^{(n)}}$ represents the number of features for population $p^{(n)}$ of cell $n$ (complete notation provided in Table A1).

**Spatial locality of interactions.** Interactions are assumed to be local in space; hence we restrict attention to a radius $d_{n,n'}(t) \leq R$ (Fig. 1 top).

**Feature space learning.** Cellular features are transiently measurable; for example, certain cells may only be visible in a subset of frames due to occlusion, movement, or other limitations Fig. S6. These properties make accurate long-term tracking an ongoing challenge (Maška et al., 2023). Thus, it is desirable to avoid reliance on error-prone global trajectories and instead focus on short-horizon correspondence–predicting targets at time $t$ are from their spatial neighbors at time $t-1$. Cellular identity and long-term relations are then preserved by operating in feature space–learned rules generalize across cells with similar states (Fig. 1 middle).

**Learning objective.** We seek to recover biologically meaningful, time-varying interaction rules from transient multi-population observations, avoiding fixed cell identities, towards revealing the mechanisms through which cellular networks govern biological functions and disease processes.

**Balancing predictive accuracy with interpretability.** We seek a model that not only maximizes fit but also maintains parsimony through: (i) limiting the number of learned parameters, and (ii) providing components that directly map to observable space, enabling mechanistic interpretation of interactions.

## 4 THE LICCHIE APPROACH

**Feature-evolution model.** Looking to identify how multi-cellular interactions drive changes in cell-states, we describe interactions via feature evolution–modeling the state vector $\boldsymbol{x}_t^{(n)}$ of cell $n$ at time $t$ as the weighted sum of learnable interaction matrices capturing influences from previous time-point, namely,

$$\boldsymbol{x}_t^{(n)} = \sum_{k=1}^N \mathbf{1}_{d_{n,k}(t) \leq R} d_{n,k}(t) \boldsymbol{A}_t^{(n,k)} \boldsymbol{x}_{t-1}^{(k)}. \tag{1}$$

For clarity of presentation, we hereafter omit the distance-driven reweighting. Notably, in the above formulation feature-level effects are captured by linear relations. This approach was deliberately chosen, prioritizing direct interpretability of the results–deriving understandable interaction rules over model complexity, and justified via empirical validations (App. A). Importantly, the above formulation enforces temporal consistency, requiring reconstruction of cell identity at time $t$ based on its state and local neighborhood at $t-1$. Together with the model design choices, described shortly, this construction ensures that the interaction matrices depict genuine cell-cell dynamics, decoupling them from transient noisy events which may be captured in the data.

**The interaction matrices.** A matrix $\boldsymbol{A}_t^{(n,k)} \in \mathbb{R}^{m \times m}$ represents the effect of source cell $k$ on target cell $n$ at time $t$, conditioned on their previous states. An entry $\left[\boldsymbol{A}_t^{(n,k)}\right]_{i,j}$ captures the influence of feature $j$ of cell $k$ on feature $i$ of cell $n$, from $t-1$ to $t$ (e.g., how much the size, i.e., feature $i$ of cell $n$ at time $t$, was impacted by the position, i.e., feature $j$, of cell $k$, at time $t-1$, Fig. 1 middle). Here, we assume that all cells share the same feature set, yet the model naturally extends to population-specific features (App. B).

**Cross-interaction similarity.** Naturally, pairs with similar stacked state vectors induce similar interaction maps. With that, if the source vector $\boldsymbol{x}_t^{(k)}$ lies in the null space of a matrix $\widetilde{\boldsymbol{A}}$, then $\boldsymbol{A}_t^{(n,k)} + \widetilde{\boldsymbol{A}}$ is an equivalent solution to $\boldsymbol{A}_t^{(n,k)}$ in terms of fit—thus applying regularization can improve inference by enforcing consistency and smoothness across parameter changes (App. C). We employ this by constraining distances between $\boldsymbol{A}_t^{(n,k)}$ with the respective distance in feature space,

$$\|\boldsymbol{A}_t^{(n,k)} - \boldsymbol{A}_t^{(n',k')}\|_2^2 \le \epsilon\big(\delta_{\boldsymbol{A}_t^{(n,k)},\boldsymbol{A}_t^{(n',k')}}\big), \tag{2}$$

where $\varepsilon(\cdot)$ decreases with the distance $\delta_{\boldsymbol{A}_t^{(n,k)},\boldsymbol{A}_t^{(n',k')}}(t)$ between the stacked state vectors of the two pairs at time $t$. In practice implemented as a *soft, distance-weighted penalty* on interaction matrices distances.

**Low-rank interaction-features regularization.** To reflect a limited number of shared biological processes, we impose a low-rank structure by expressing each $\boldsymbol{A}_t^{(n,k)}$ via a restricted set of global rank-1 components with pair-/time-specific weights,

$$\boldsymbol{A}_t^{(n,k)} = \sum_{j=1}^{J} W_{j,t}^{(n,k)} \boldsymbol{c}_j \boldsymbol{r}_j^\top = \sum_{j=1}^{J} W_{j,t}^{(n,k)} \underbrace{\boldsymbol{M}_j}_{\boldsymbol{c}_j \boldsymbol{r}_j^\top} \tag{3}$$

where $\{\boldsymbol{c}_j\}_{j=1}^J$ and $\{\boldsymbol{r}_j\}_{j=1}^J$ are the set of underlying components such that each $\boldsymbol{c}_j \in \mathbb{R}^m$ and $\boldsymbol{r}_j \in \mathbb{R}^m$, spans each $\{\boldsymbol{A}_t^{(n,k)}\}_{t,n,k}$, and each term $\boldsymbol{M}_j := \boldsymbol{c}_j \boldsymbol{r}_j^\top$ is a rank-1 matrix constructed via the outer product of two $m$-dimensional vectors. Each $\boldsymbol{W}^{(n,k)} \in \mathbb{R}^{J \times T}$ is the corresponding sparse per time-point weight matrix, such that $W_{j,t}^{(n,k)}$ is the contribution of components $\boldsymbol{c}_j, \boldsymbol{r}_j$ to the interaction from $k$ to $n$ at time $t$ (Fig. 1 bottom).

**The multi-feature vector.** While the set of observed features is defined by the data in hand, the induced multi-feature vector, $x_t^{(n)} \in \mathbb{R}^{m_{p(n)}}$, can be defined based on the task in hand and vary from engineered, pre-defined, interpretable features based on observed attributes (e.g. spatial, kinematic and available phenotypic data, Stirling et al. (2021)) to a latent space embedding of the observation (e.g. a deep learning representation, Moshkov et al. (2024)). In what follows, we focus on the former, as it allows for direct interpretation of observed feature relations. Importantly, when considering the latter, the setting can uncover dependencies and importance of different latent dimensions, desirable for explainability of AI models (Saranya & Subhashini, 2023).

**Learning low-rank interactions via optimization.** Balancing fidelity with the assumptions above, we solve:

$$\mathcal{L} = \sum_{n=1}^{N} \sum_{k=1}^{N} \left[ \|\boldsymbol{x}_t^{(n)} - (\mathbf{1}_{d_{n,k}(t) \leq R}) \boldsymbol{A}_t^{(n,k)} \boldsymbol{x}_{t-1}^{(k)}\|_2^2 + \lambda_1 \|\boldsymbol{A}_t^{(n,k)} - \sum_{j=1}^{J} W_{j,t}^{(n,k)} \boldsymbol{c}_j \boldsymbol{r}_j^T \|_F^2 \right. \quad (4)$$

$$+ \lambda_2 \sum_{\substack{\boldsymbol{A}' \in \{\boldsymbol{A}_{t'}^{(n',k')}\}_{n',k',t'} \\ \boldsymbol{A}' \neq \boldsymbol{A}_t^{(n,k)}}} \left[ \|(\mathbf{1}_{d_{n,k}(t) \leq R}) e^{-\sigma \delta(\boldsymbol{A}_t^{(n,k)}, \boldsymbol{A}')} (\boldsymbol{A}_t^{(n,k)} - \boldsymbol{A}')\|_F^2 \right]$$

$$\left. + \lambda_3 \|\text{vec}(\boldsymbol{A}_t^{(n,k)})\|_1 + \lambda_4 \|\boldsymbol{W}_{:,t}^{(n,k)}\|_1 \right],$$

where $R$ is the interaction radius, $\sigma$ is a parameter controlling the decay of similarity with distance in feature space, $\lambda_1, \lambda_2, \lambda_3, \lambda_4$ are scalar regularization weights that balance the different terms. $\boldsymbol{A}_t^{(n,k)}, \boldsymbol{A}'$ denote interaction matrices corresponding to feature pairs such that $\delta(\boldsymbol{A}_t^{(n,k)}, \boldsymbol{A}')$ is the distance between these pairs in the feature space. The first term ensures fidelity to the observed dynamics by minimizing prediction error, the second term enforces a low-rank structure by approximating each $\boldsymbol{A}$ as a weighted sum of shared components, the third term encourages smoothness by penalizing differences between nearby interactions in feature space, and the two final terms impose sparsity on the interactions and corresponding weights (see discussion on hyper-parameters in App. D).

We fit the model using an iterative approach, iterating between estimating the interaction matrices $\{\boldsymbol{A}_t^{(n,k)}\}$ via LASSO Tibshirani (1996), the shared components $\{\boldsymbol{c}_j, \boldsymbol{r}_j\}$, and their weights $\{\boldsymbol{W}_{:t}^{(n,k)}\}$ using PARAFAC (Harshman et al., 1970), alternating between the following steps.

1. **Update interaction matrices $\boldsymbol{A}$:** Given the current components and weights, infer each interaction matrix by minimizing the reconstruction and smoothness losses (for notation clarity, we denote in the equation $\widehat{\boldsymbol{A}}_t^{(n,k)} := \boldsymbol{A}$)

$$\widehat{\boldsymbol{A}}_t^{(n,k)} = \arg\min_{\boldsymbol{A}} \|\boldsymbol{x}_t^{(n)} - \sum_{k=1}^{N} (\mathbf{1}_{d_{n,k}(t) \leq R}) \boldsymbol{A} \boldsymbol{x}_{t-1}^{(k)}\|_2^2 + \lambda_1 \left\| \boldsymbol{A} - \sum_j W_{j,t} \boldsymbol{c}_j \boldsymbol{r}_j^\top \right\|_2^2$$

$$+ \lambda_2 \sum_{\boldsymbol{A}' \neq \boldsymbol{A}} e^{-\sigma \delta(\boldsymbol{A}, \boldsymbol{A}')} \|\boldsymbol{A} - \boldsymbol{A}'\|_2^2 + \lambda_3 \|\text{vec}(\boldsymbol{A})\|_1 \quad (5)$$

where $\lambda_1, \lambda_2,$ and $\lambda_3$ are hyper-parameters balancing low-rank fidelity, interaction smoothness, and sparsity level.

2. **Update shared components $\boldsymbol{c}_j, \boldsymbol{r}_j$:** With fixed weights and $\{\boldsymbol{A}_t^{(n,k)}\}$, optimize the low-rank factors to best approximate the interaction matrices across all pairs and times:

$$\{\widehat{\boldsymbol{c}}_j, \widehat{\boldsymbol{r}}_j\}_{j=1}^{J} = \arg\min_{\{\boldsymbol{c}_j, \boldsymbol{r}_j\}} \sum_{n,k,t} \left\| \boldsymbol{A}_t^{(n,k)} - \sum_{j=1}^{J} W_{j,t}^{(n,k)} \boldsymbol{c}_j \boldsymbol{r}_j^\top \right\|_2^2 \quad (6)$$

3. **Update weights $\forall t : \{\widehat{\boldsymbol{W}}_{:,t}^{(n,k)}\}_{n,k}$:** Given the current components and $\{\boldsymbol{A}_t^{(n,k)}\}$, update the weights for each time step by minimizing the reconstruction error with sparsity constraints:

$$\widehat{\boldsymbol{W}}_{:,t}^{(n,k)} = \arg\min_{\boldsymbol{W}_{:t}^{(n,k)}} \left\| \boldsymbol{A}_t^{(n,k)} - \sum_{j=1}^{J} W_{j,t}^{(n,k)} \boldsymbol{c}_j \boldsymbol{r}_j^\top \right\|_2^2 + \lambda_4 \|\boldsymbol{W}_{:,t}^{(n,k)}\|_1 \quad (7)$$

where $\lambda_4$ controls the sparsity of the number of active components in the $k \to n$ interaction at time $t$.

LICCHIE can thus identify interactions between individual cells based on their features (i.e., with good resolution and not mean-field) while considering all cells within the interaction radius, thus providing a coherent and unified representation in the feature space. Notably, environmental effects that are not caused by other cells, as well as the natural evolution of a cell, can be captured via the self–self interaction matrix of a cell with itself and integrated into the summation of all interactions, which enables disentangling these effects. The steps are summarized in Algorithm 1 and Fig. 1, optimization details and complexity analysis are provided in App. E.

---

**Algorithm 1** LICCHIE: Low-rank interpretable cell-cell hidden interactions via embedding model

---

**Require:** Observed cell features $\{\boldsymbol{x}_t^{(n)}\}$, interaction radius $R$, number of components $J$, regularization weights $\lambda_1, \lambda_2, \lambda_3$, feature-space decay $\sigma$
**Ensure:** Interaction matrices $\{\boldsymbol{A}_t^{(n,k)}\}$, shared components $\{\boldsymbol{c}_j, \boldsymbol{r}_j\}$, weights $\boldsymbol{W}$
  1: Initialize $\boldsymbol{A}_t^{(n,k)}, \boldsymbol{c}_j, \boldsymbol{r}_j, \{\boldsymbol{W}_{:t}^{(n,k)}\}_{n,k,t}$ randomly from normal *i.i.d* distribution
  2: **repeat**
  3:     **Update interaction matrices** $\{\boldsymbol{A}_t^{(n,k)}\}$**:**
  4:     **for** each cell pair $(n, k)$ and time $t$ **do**
  5:         Infer $\boldsymbol{A}_t^{(n,k)}$                    ▷ equation 5, balances low-rank, smoothness, and sparsity
  6:     **end for**
  7:     **Update shared components** $\boldsymbol{c}_j, \boldsymbol{r}_j$                    ▷ equation 6
  8:     **Update weights** $\{\boldsymbol{W}_{:t}^{(n,k)}\}_{n,k,t}$                    ▷ equation 7
  9: **until** convergence or maximum iterations reached
 10: **Return** $\{\boldsymbol{A}_t^{(n,k)}\}_{n,k,t}, \{\boldsymbol{c}_j, \boldsymbol{r}_j\}_j, \{\boldsymbol{W}_{:t}^{(n,k)}\}_{n,k,t}$

---

## 5 EXPERIMENTS

### 5.1 LICCHIE RECOVERS TRUE COMPONENTS IN SYNTHETIC DATA

**Setup.** We simulate two populations (25 and 10 cells, 10 features each) evolving over $T$=50 frames (Fig. 2a). Eight ($J = 8$) rank-1 ground-truth motifs $\{M_j\}_{j=1}^8$ were generated to be orthogonal under the Frobenius inner product, with per-pair, per-time weights that are 20% sparse. Initial features came from population-specific Gaussians; cell motion followed trigonometric trajectories with population-specific amplitudes and frequencies; Gaussian noise was added each step. Edges were constructed by $k$NN ($k = 10$) on $t_0$ features and pruned per frame by a spatial radius $R$. At each time, targets were updated by a weighted sum of sources within the radius. To prevent divergence, we scaled updates to keep the spectral radius below 1.

**Baselines.** We benchmark LICCHIE against four alternative methods spanning linear, switching–LDS, and GNN approaches: (i) **Global linear dynamics**: a single time-fixed matrix $\boldsymbol{A}$ fit by ordinary least squares (OLS) on all cells to predict $\boldsymbol{x}_{t+1}$ from $\boldsymbol{x}_t$; (ii) **Per-target linear dynamics**: one OLS model per target cell $n$, yielding $\boldsymbol{A}_n$; (iii) **Per-cell SLDS**: a single switching linear dynamical system fit per trajectory; and (iv) **GNN**: a distance-weighted spatial graph trained to predict $\boldsymbol{x}_{t+1}$ from $\boldsymbol{x}_t$. (App. H). Methods (i)–(ii) produce explicit interaction matrices that enable comparison with LICCHIE's output, yet are limited in their capacity to capture complex interactions that vary in space and time. Methods (iii)–(iv) lack comparable interpretable components (App. H). This limitation also prevents comparison to alternative deep learning methods.

**Results.** We first validate LICCHIE's inferred interactions via their predictive power; we observe strong performance in prediction of cell-type identity of either source (0.86) or target (0.87) cells, or both jointly (0.83, Fig. 2c). Next, in comparison to linear baselines (methods i and ii), LICCHIE produced interaction structures closer to the ground truth construction $\{M_j\}_{j=1}^8$ (Fig. 2d). Visualization of the learned components from the per-cell SLDS (method iii, Fig. S3) and the GNN (method iv, Fig. S4) depicts that these approaches are ill-suited for the targeted task, with the former constrained by abrupt switching transitions (Fig. S3a), while the latter requires additional processing to approximate transitions via uninterpretable dense loadings (Fig. S4b,c). At last, LICCHIE

obtained the most accurate predictions for the interaction matrices, compared to all baselines, considering the reconstruction error (Fig. 2e,f) and correlation with ground-truth (Fig. S2).

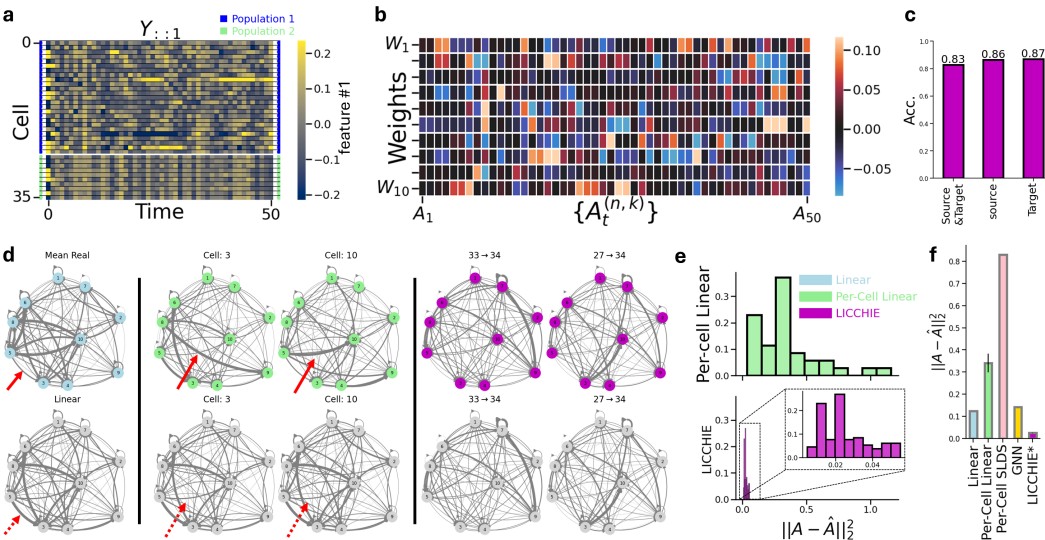

Figure 2: *Synthetic data overview and results.* **a**, Examples of generated features illustrating population differences (blue/green side bars). **b**, Ground-truth weights show sparse structure. **c**, Accuracy results for LICCHIE's cell type prediction using the inferred interactions (logistic regression with 5-fold cross-validation); predictions of source and target (0.83), or source (0.86) and target (0.87) independently. **d**, Rank-1 motif inference (top row; left to right: linear, per-cell linear, LICCHIE) vs. ground truth (bottom row). Red arrows highlight discrepancies between ground truth (dotted) and baseline-identified networks (solid). **e**, Per-pair MSE distributions vs. ground truth for the per-cell linear baseline (top) and LICCHIE (bottom). **f**, Average MSE for the global time-fixed linear baseline, per-cell dynamics, per-cell SLDS, GNN, and LICCHIE (additional metrics in Fig. S2)

## 5.2 LICCHIE UNCOVERS INTERPRETABLE TUMOR−NK INTERACTIONS

**Data and biological relevance.** As our real-world application, we analyze patient-derived gastric tumor organoids co-cultured with primary human NK cells in 3D time-lapse confocal imaging (8–12 h, 4-min cadence, 27–30 z-slices per frame; Liu et al. (2024)). Organoid co-cultures provide an established platform to study interactions between motile immune effectors and tumor cells in a controlled 3D microenvironment (Polak et al., 2024). These platforms allow investigation of contact-resolved immune synapses where brief, often serial engagement events underlie cytotoxic action (Vanherberghen et al., 2013; Dekkers et al., 2023) and imaging readouts from such co-culture conditions have proven translational value (Alieva et al., 2023). We hence apply LICCHIE to study time-varying, local influences between tumor-immune co-cultures.

**The extracted features.** As noted by Liu et al. (2024), cells in the data presenting variability in cell morphology and intensity distribution (Fig. 3a). Hence, we extracted spatial, morphology, and intensity features, relying on cell-level segmentation masks from a representative experiment. We normalize the features such that each feature, at population level (NK or tumor cells), is normally distributed, $f_i^{(p)} \sim \mathcal{N}(\mu = 0, \sigma = 1)$, implying that all features, although representing geometric properties, attain negative values (App. F, Table A2). We fitted LICCHIE with $J=10$ components.

**Components and structure.** LICCHIE identified components that reveal diverse interaction patterns, including *localized targets* (e.g., $M_2, M_4$), *source-emerging* (e.g., $M_1, M_{10}$), self *feedback-like* interactions (e.g., $M_3$) and multi-step interactions, including direct and indirect influences (e.g., $M_6$, Fig. S7).

Interaction motifs can capture indicators of cell state, such as NK cell polarization. Upon polarization, NK cells undergo a characteristic morphological shift from round to elongated (Fig. 3a), a

process necessary for their cytotoxic activity. Analysis of motif 2 ($M_2$) and the respective weights ($W_2$) suggests that it captures predominant interactions targeting NK cells, with opposing effects depending on the source cell (Fig. 3b,c), specifically identifying that the length of the major axis (light green) of a target NK cell–a measure of its longest dimension that reflects its polarization status–is predicted by diverse source features. This suggests that in a complex co-culture system, the NK cell length (indication of its polarization) is not a singular, homogeneous event but is instead coupled to diverse source cell states.

Motif 5 ($M_5$) has a sparse interaction matrix (Fig. 3b) and predominantly captures NK-to-tumor interactions ($W_5$, Fig. 3c). This motif indicates that source cell mean intensity and sphericity, are the main predictive features of two primary target features: cell aspect ratio and a tumor cell identity indicator. Therefore, $M_5$ suggests that interaction with a subset of NK cells distinguished by their brightness and sphericity is indicative for tumor-cell elongation. Interestingly, motif 6 ($M_6$) also centers on cell aspect ratio through different means, including direct and indirect influences of several features. For example, cell sphericity (light green) directly and indirectly through cell height (gray) influences cell aspect ratio. This underscores LICCHIE's ability to disentangle multiple processes with overlapping actors while distinguishing the underlying mechanisms.

Lastly, motif 7 ($M_7$) highlights the centrality of source cell size in shaping multiple target features, and the respective weights ($W_7$) indicate that $M_7$ mostly negatively regulates tumor-to-NK interactions. A particularly notable target feature is cell surface area. In NK cells, an increased surface area could indicate the formation of a lytic immunological synapse (Orange, 2008). This may indicate an impact of interactions between tumor cells of specific size and the process of NK-cell spreading prior to synapse formation.

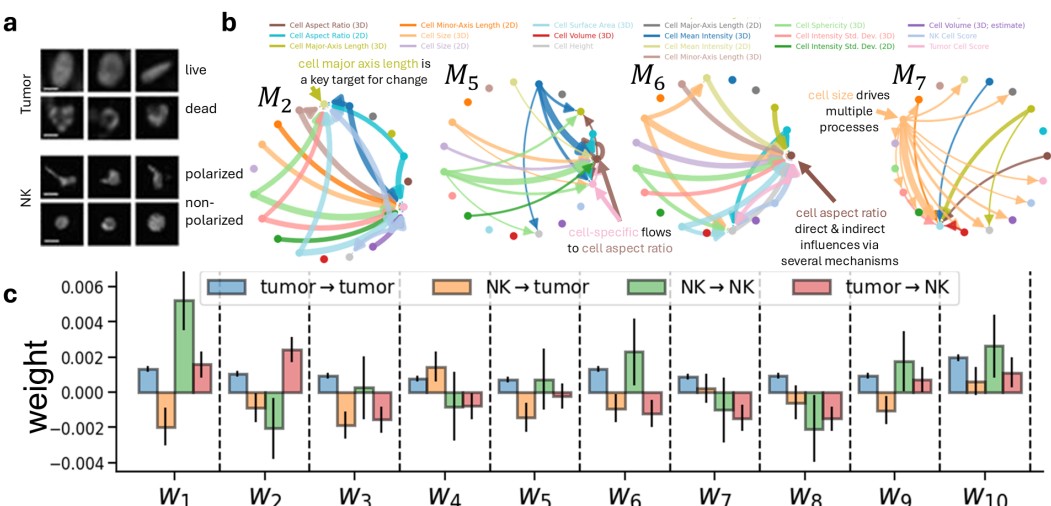

Figure 3: *LICCHIE identifies meaningful cellular interactions in real-world data*. **a,** Representative images illustrate morphology/intensity differences between phenotypes (Liu et al., 2024). **b,** Examples of LICCHIE's rank-1 motifs ($J$=10; full set in Fig. S7). **c,** Component weights by interaction class (tumor→tumor, NK→NK, NK→tumor, tumor→NK) show class-specific activation patterns.

**Interaction-type specificity.** Component activations differ across interaction classes. As shown in Fig. 3c, components display class-specific activation patterns across tumor→tumor, NK→NK, NK→tumor, and tumor→NK pairs. Interesting relations observed include:

- *Opposing weights between cell classes*. Component 6 (weighted by $W_6$) presents positive weights for same-cell interactions and negative weights for opposing cell classes. Mechanistically, similar shapes may promote symmetric contact geometry or aligned polarity, resulting in higher positive $w_6$ values for same-cell pairs, whereas mismatched shapes reduce contact or produce asymmetry, leading to negative $W_6$ values for different-cell pairs.
- *Single interaction dominance*. Components 1 and 6 ($W_1$, $W_6$), show clear dominance of a single interaction type (e.g., NK → NK), which may reflect specialized functional role.

- *Source disentangling*. Components 1 and 2 ($W_1$, $W_2$) capture predominant interactions targeting tumor and NK cells respectively, with opposing weight signs given the source.
- *Cell-specific regulation*. Component 3 ($W_3$) is mainly associated with interactions between tumor and NK cells (rather than homogeneous cell-type interactions.

### 5.3 FROM MOTIFS TO PHENOTYPES: USING ENGINEERED CELLS FOR PREDICTION

**Weight sweeps on unseen data.** A unique property of LICCHIE is the globality of the rank-1 components, $\{M_j\}_{j=1}^{J}$ that enables generalization to unseen data that can drive critical biological downstream processes. We assessed whether individual components are associated with increased NK polarization or tumor-cell death by modulating each component's weight while keeping source features within the observed feature space.

**Procedure.** For each $M_j$, $j = 1 \ldots J$ component, we gradually increased its weight from $0$ to $2$ in $20$ steps and applied it to various combinations of source features. We then assigned each inferred target a polarization score (for NK cells) or a death score (for tumor cells) using a distance-weighted $k$-nearest neighbor classifier ($k=150$, total # samples from all time points = $20,437$) based on Euclidean distance in feature space and normalized by phenotype frequency (Fig. 4a).

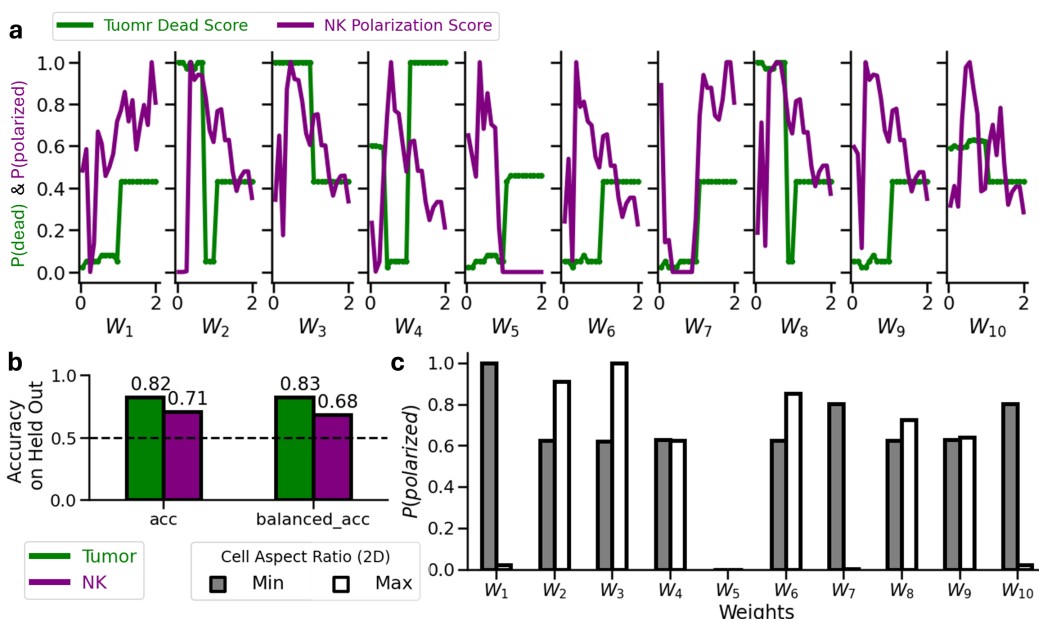

Figure 4: *Probing phenotype effects via weight modulation*. **a**, Gradually modulating the weights on unseen data points reveals diverse encoding mechanisms for NK polarization (magenta) and tumor cell death (green). **b**, Accuracy and balanced-accuracy for phenotype prediction using inferred component weights on held-out data. **c**, Effect of extreme source-feature values on target NK polarization.

**Predictive evaluation against baselines.** The biological relevance of models' outputs is tested by assessing their predictive power to known biological properties. Comparing LICCHIE to two baseline methods (SLDS and GNN; Sec. 5.1), we evaluate accuracy of cell type prediction, interaction types, and phenotypes, directly from the interaction matrices. These tests demonstrate that LICCHIE is the only method whose outputs could directly be used to obtain meaningful predictions (App. I, Fig. S8, Fig. S9, Fig. S10).

**Predictions from inferred weights.** Using weights inferred by LICCHIEon unseen frames, we trained a cross-validation logistic regression to predict target phenotypes. Although phenotypes are not used during training, weight-only features predicted phenotypes on held-out data (Fig. 4b), indicating that component activations capture phenotype-relevant interaction signals.

**Cell status relations in interactions.** Focusing on LICCHIE's interpretable components we notice they yield distinct death/polarization score trajectories with non-linear changes as weights increased (Fig. 4a). For tumor-cell death, some components (e.g., $M_1, M_5, M_6$, Fig. S7) showed increasing scores with weight, whereas others (e.g., $M_2, M_3, M_8$) showed decreases, often most prominently around weight $\approx 1$. Overall, tumor death scores exhibited phase-transition-like effects (non-smooth changes), consistent with the acute nature of cell death. NK-cell polarization trends were more variable; for example, $M_7$ increased polarization at higher weights, while $M_5$ decreased it. In some cases (e.g., $M_5$), tumor-death and NK-polarization scores moved in opposite directions as weight increased.

**Source-feature extremes.** We also examined the impact of source features on NK polarization under maximal activation ($W=2$). For each source feature, we compared scores at its empirical minimum vs. maximum (others fixed at their means). We found that source properties give rise to both diversity and correlations in the potential effect of each component. For example, when component $M_1$ is applied to a source feature vector with a minimized cell aspect ratio, it increases the polarization score of the target. In contrast, when applied to the maximum cell aspect ratio, the target cell's polarization score remains near zero. Other components, such as $M_4$ or $M_9$, show similar target polarization scores under both minimized and maximized cell aspect ratio values, suggesting that this feature may not be critical in these circuits for driving changes in target polarization (Fig. 4c). Altogether, these results highlight the potential of LICCHIE to guide the design of future interventions when applied to additional datasets.

## 6 DISCUSSION AND FUTURE WORK

We presented LICCHIE, an interpretable method to study cell-cell interactions capturing modular, temporally consistent, feature-specific cell–cell interactions. The PARAFAC rank-1 decomposition yields interpretable source–target components, making it explicit which sources influence which targets rather than obscuring them in complex, opaque patterns. The cell state feature representations are user defined; allowing for problem specific optimization. Over these, LICCHIE reveals interpretable motifs capturing features role as targets or sources within modular cell–cell interactions through coordinated, overlapping, and co-modulatory effects. LICCHIE's generality enables use across experimental systems—from isolated cells in a dish to complex 3D models—capturing the distinct cell-cell communication aspects they exhibit. The observed system shall guide the interpretations of the inferred interaction motifs. Applied to live-cell imaging data of tumor–NK co-cultures, LICCHIE revealed key patterns of cellular communication–identified components with distinct feature relations–revealing the modular biological regulation that integrates diverse features into phenotypic outcomes. An important advantage of LICCHIE is that it can address the challenge of modeling interactions from separate, asynchronous measurements while preserving data-specific details. Its global vectors $c_j$ and $r_j$ allow new data to be analyzed quickly by reusing the components and adjusting only the weights, avoiding full-model retraining. Notably, LICCHIE naturally handles data imbalance by its ability to re-weight interactions based on the prevalence of each population and interaction type, ensuring adequate representation for all interactions.

**Limitations and future directions.** The model assumes linear or additive effects, which may not fully capture certain non-linear dynamics; this design choice promotes interpretability, and found valid in practice through reconstruction accuracy (see discussion in App. A). Future work can incorporate non-linear activation functions $f(\cdot)$ to each interaction pair to improve flexibility. Here, interactions beyond the defined radius are not captured; this limitation can be easily addressed by applying a kernel and sampling from an appropriate probability distribution that varies with distance. Further biological interpretation of components requires additional supervision via experimental data that can be obtained by applying LICCHIE to additional systems with dedicated measurements. In addition, future experimental work coupled with LICCHIE could establish the mechanisms and causal relationships underlying the identified interaction motifs, for example through targeted perturbations along effector–target axes. Moreover, by isolating the morphological features underlying such interaction motifs LICCHIE provides a framework to link dynamic imaging phenotypes with molecular information by layering in spatial and single-cell multi-omics in subsequent studies. Lastly, using temporal mapping tools (e.g., optimal transport, Klein et al. (2025)), LICCHIE can be applied to spatio-temporal single-cell datasets, providing insights into interactions over a richer cell-state representation space, which may be more directly applicable for therapeutic applications.

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

# Appendix

## A  ON THE LINEARITY ASSUMPTION

Linear approximations are widely used in dynamics because they simplify analysis and control. Around an equilibrium point or operating region, many nonlinear systems can be locally approximated with linear models (i.e., via Taylor expansion). Accordingly, locally linear models, including linear state-space models, auto-regressive models, GLMs, and linearized ODEs, are standard for modeling, e.g., movement (Patterson et al., 2008), population dynamics (Boling Jr, 1973), and diverse biological interactions (Brauer & Kribs, 2016; Mudrik et al., 2025; Andersson et al., 2005).

A unique advantage of maintaining a linear transition matrix $A$ in biological systems is interpretability: in our framework each entry $A_{ij}$ directly reflects the effect of feature $i$ on feature $j$, and can be linked to the observation space. LICCHIE extends typical linear systems model to be flexible distance-varying time-changing cell-specific locally changing dynamics, and thus is expressive enough to capture the complexity of temporally evolving cell–cell interactions, while preserving this interpretability: each entry in a rank-1 component corresponds to a source–target effect between cells.

To account for non-linear dynamics one can consider non-linear activations, e.g. $x_t = \sigma(A_t)x_{t-1}$. However, including these prevents direct interpretation of the results, necessary for biological discovery.

## B  POPULATION-SPECIFIC FEATURE SPACE

The LICCHIE framework can naturally extend to account for multiple cell populations exhibiting distinct feature subsets. To do so, the fixed component sets $\{c_j\}$ and $\{r_j\}$ are replaced by population-specific sets $\{c_j^{(p)}\}_{j=1}^J$ and $\{r_j^{(p)}\}_{j=1}^J$, where $p$ indexes the population. Within each set, the vector dimensions are consistent, but they may differ between populations.

For within-population interactions (e.g., population $p$), the $A$ matrices are modeled using the components of the corresponding $p$ set. For cross-population interactions, the sets are chosen according to the participating populations: for an interaction from a cell in population $p$ to a cell in population $p'$, we use $\{r_j\}$ from $p$ and $\{c_j\}$ from $p'$, and vice versa. The resulting interaction matrix may not be square if the number of features differs between populations.

Notably, the above extension preserves the interpretability afforded by the component structure while allowing flexibility across nuanced population dynamics.

## C  AVOIDING EQUIVALENT SOLUTIONS THROUGH DISTANCE CONSTRAINTS

Learning interactions for all feature and spatial combinations is intractable and highly non-interpretable for scientific purposes. Hence, we leverage our assumption of cross-interaction similarity in $A$ matrices that represent similar source-target feature distributions.

When fitting dynamics, similarities in source and target vector pairs should ideally result in similar transition matrices (i.e., if $(n, k)$ is close to $(n', k')$, then their interactions $A_t^{(n,k)}$ and $A_t^{(n',k')}$ should be similar). Yet, when fitting dynamics in an unconstrained way, if the source vector $x_t^k$ lies in the null space of any matrix $\widetilde{A}$, then $A_t^{(n,k)} + \widetilde{A}$ is an equivalent solution to $A_t^{(n,k)}$ in terms of fit. This means that we must identify a solution that is consistent across pair, enforcing smoothness across parameter changes.

Hence, we introduced the distance constraint; forcing distances between $\{A\}$ matrices to follow distances in the corresponding feature space, i.e.,

$$\|A_t^{(n,k)} - A_t^{(n',k')}\|_2^2 \leq \epsilon\big(\delta_{(n,k),(n',k')}(t)\big)$$

where $\epsilon$ is some function that depends on $\delta_{(n,k),(n',k')}(t)$ refers to the distance between stacked feature vectors of interaction pairs $(n, k)$ and $(n', k')$ at time $t$.

## D OBJECTIVE HYPER-PARAMETERS

The LICCHIE objective includes a minimal set of hyper-parameters introduced to balance representation sparsity ($\lambda_3$, $\lambda_4$), smoothness ($\lambda_2$), data fidelity, and low-rank approximation ($\lambda_1$). In addition, the framework provides the user with the flexibility to set the maximum rank of each interaction, $J$, and the effective radius of interaction $R$; both are quantities that should be set with respect to the characteristics of the biological system studied, i.e., accounting for the total number of features and the representative length scale.

In practice, hyper-parameters can be tuned by running a parameter search (e.g., grid search) while optimizing an information criterion that balances degrees of freedom and model complexity, such as AIC (Akaike Information Criterion) or BIC (Bayesian Information Criterion, Chakrabarti & Ghosh (2011)).We provide guidance and further intuition for tuning these,

- **Low-rank constraint** ($J$, $\lambda_1$): These control how strongly the $\boldsymbol{A}$'s follow the low-rank approximation. When $J$ is larger, or when $\boldsymbol{A}$ is naturally close to low-rank, this term should converge to 0, and the sensitivity to $\lambda_1$ will be minimal (since its multiplier is close to zero). For highly complex, high-dimensional, and fast-varying dynamics, the low-rank Frobenius norm may not be small. In such cases, we recommend monitoring the rank-1 reconstruction over iterations (calculated automatically within the model). If the low-rank reconstruction remains low across iterations, users should increase both $J$ and $\lambda_1$.

- **Interactions smoothness** ($\lambda_2$): The smoothness of interactions in feature space, mainly preventing null-space solutions from causing abrupt switches (see App. C), is controlled by $\lambda_2$. In practice, a small value $\lambda_2 < 1$ is usually sufficient. We recommend examining the smoothness of interactions under gradually changing parameters to confirm that the identified interactions align with biological or system-specific assumptions.

- **Sparsity constraint** ($\lambda_3$, $\lambda_4$): Sparsity is induced by these by modulating the number of zero or near-zero entries. $\lambda_3$ promotes sparsity in the interaction matrices, facilitating a clearer understanding of the functionality of each interaction. $\lambda_4$ encourages interactions to be composed a limited number of $\{\boldsymbol{c}_j\}$ and $\{\boldsymbol{r}_j\}$ components, valuable for,

  1. Understanding the modular role of each $\{\boldsymbol{c}_j, \boldsymbol{r}_j\}$ pair in driving the overall interaction,
  2. Identifying commonalities and differences between interaction types (if all components are used in all interactions, distinguishing differences is harder), and
  3. Reducing noise captured by other components, yielding a more robust solution.

- **Effective interaction range** ($R$): The radius, $R$, defines a disk around a cell $i$ in which interaction are considered, i.e., cells within this disk are considered interacting with it. This shall be set with respect to the system's length scales and desired scope of interactions one wishes to study. e.g., one can choose to limit the analysis to first-order interactions, accounting for cells in immediate neighborhood or larger value for long-range effects.

## E OPTIMIZATION AND COMPLEXITY

To fit the model's components we use LASSO (Tibshirani, 1996), estimating the interaction matrices, $\{\boldsymbol{A}_t^{(n,k)}\}$, and Parallel Factor Analysis (PARAFAC, Harshman et al. (1970)) for the shared components $\{\boldsymbol{c}_j, \boldsymbol{r}_j\}$, and their weights $\{\boldsymbol{W}_{:t}^{(n,k)}\}$. PARAFAC is a method to decompose high-dimensional tensor data (multi-way) into underlying, independent components, extending the Principal Component Analysis (PCA) model to more than two dimensions. This provides a unique solution that allows for the recovery of pure components.

The complexity of the optimization is as follows; let $N$ denote the number of cells, $T$ the number of time points, $d$ the number of features, $k$ the average number of neighbors (within radius), $S = k - 1$ the number of sources, $R$ the number of outer iterations, $C$ the number of PARAFAC components, $I$ the number of PARAFAC ALS iterations, $M$ the number of past $A$'s maintained for similarity search, and $E \approx T \cdot N \cdot k$ the total number of interactions.

**Pre-processing (neighbors within radius).**

- Naive all-pairs distances (per time): $O(T \cdot N^2)$ time, $O(1)$ extra memory.

- With a spatial index (grid/kd-tree): build + query costs $O(T \cdot N \log N + T \cdot E)$ time. The neighbor graph requires $O(E)$ memory.

**Per target per time (one cell as target).**

- Build wide $\boldsymbol{A} \in \mathbb{R}^{d \times (S \cdot d)}$: $O(S \cdot d^2)$.
- Least-squares solve $x_{\text{target}} = \boldsymbol{A}\, x_{\text{sources}}$: $O(S^2 \cdot d^3)$ time, $O(S \cdot d^2)$ memory.
- Maintain $M$ most similar past $\boldsymbol{A}$'s: similarity search over history $H \approx R$, costing $O(H \cdot S \cdot d^2)$ (naive).
- Decompose each $A$ (reshaped as a tensor $d \times S \times d$) with PARAFAC/CP: ALS per $\boldsymbol{A}$ costs $O(I \cdot C \cdot S \cdot d^2)$ time, $O(C \cdot (d + S + d))$ memory.

**Totals per outer iteration (across all targets and times; $\sim T \cdot N$ problems).**

- LS solves: $O(T \cdot N \cdot S^2 \cdot d^3)$.
- Similarity searches: $O(T \cdot N \cdot R \cdot S \cdot d^2)$ (if $H \approx R$).
- PARAFAC: $O(T \cdot N \cdot I \cdot C \cdot S \cdot d^2)$.

**Overall complexity.**

$$O\big(T \cdot N \cdot \big(S^2 d^3 + (R + I \cdot C)Sd^2 + \log N + k\big)\big).$$

## F  THE TUMOR-NK DATASET

We used publicly available data from (Liu et al., 2024), accessed via Zenodo. The complete dataset includes 3D live-cell imaging datasets of gastric tumor organoids co-cultured with primary human Natural Killer (NK) cells. The dataset includes multiple sessions recorded asynchronously under varying experimental conditions (see (Liu et al., 2024) for more details). We selected a representative random session, 'GX048-TO + NK cell (IL-15)', in which NK cells were stimulated with IL-15.

## G  TABLES

Table A1: Summary of defined notations.

| Notation | Meaning |
|---|---|
| $N$ | Number of interacting cells |
| $T$ | Total duration of observation (number of time points) |
| $s$ | Dimensionality of the space $\mathcal{R} \subset \mathbb{R}^s$ |
| $\boldsymbol{\psi}_t^{(n)}$ | Position of cell $n$ at time $t$ in $\mathbb{R}^s$ (spatial coordinates) |
| $\boldsymbol{x}_t^{(n)}$ | State (feature) vector of cell $n$ at time $t$ in $\mathbb{R}^{m^{p^{(n)}}}$ |
| $p^{(n)}$ | Population label/type of cell $n$ |
| $\boldsymbol{P} = \{p^{(n)}\}_{n=1}^N$ | Set of population identifiers for all cells |
| $m^{p^{(n)}}$ | Number of features for population $p^{(n)}$; if constant across all cells, denote as $m$ |
| $m_{\text{spatial}}^{p^{(n)}}$ | Number of spatial features for population $p^{(n)}$ |
| $m_{\text{internal}}^{p^{(n)}}$ | Number of internal features for population $p^{(n)}$ |
| $d_{n,n'}(t)$ | Spatial distance between cells $n$ and $n'$ at time $t$, e.g., $\|\boldsymbol{r}_t^{(n)} - \boldsymbol{r}_t^{(n')}\|$ |
| $\delta_{n,n'}(t)$ | Feature distance between cells $n$ and $n'$ at time $t$, e.g., $\|\boldsymbol{x}_t^{(n)} - \boldsymbol{x}_t^{(n')}\|$ |
| $\delta_{(n,k),(n',k')}(t)$ | Feature distance between stacked state vectors of cell pairs $(n,k)$ and $(n',k')$ at time $t$. |
| $\delta_{A,A'}(t)$ | Feature distance between As |
| $R$ | Interaction radius (hyperparameter) |
| $\mathbf{1}_{d_{n,n'}(t) \leq R}$ | Indicator: 1 if distance between $n$ and $n'$ at time $t$ is $\leq R$, else 0 |

Table A2: Summary of cell features.

| Feature | Description |
|---|---|
| Cell Aspect Ratio (2D) | Ratio of major to minor axis lengths in 2D projection. |
| Cell Aspect Ratio (3D) | Ratio of principal axis lengths in 3D reconstruction. |
| Cell Major-Axis Length (2D) | Length of the longest axis in 2D projection. |
| Cell Major-Axis Length (3D) | Length of the longest axis in 3D reconstruction. |
| Cell Minor-Axis Length (2D) | Length of the shortest axis in 2D projection. |
| Cell Minor-Axis Length (3D) | Length of the shortest axis in 3D reconstruction. |
| Cell Height | Extent of the cell along the z-axis. |
| Cell Size (2D) | Area of the cell in 2D projection. |
| Cell Size (3D) | Approximate size based on 3D reconstruction. |
| Cell Surface Area (3D) | Total exposed surface area of the 3D cell. |
| Cell Volume (3D) | Computed volume of the cell in 3D. |
| Cell Volume (3D; estimate) | Estimated cell volume when full reconstruction is not available. |
| Cell Sphericity (3D) | Measure of how closely the shape approaches a sphere. |
| Cell Mean Intensity (2D) | Average intensity of the cell in 2D projection. |
| Cell Mean Intensity (3D) | Average intensity of the cell in 3D reconstruction. |
| Cell Intensity Std. Dev. (2D) | Standard deviation of pixel intensities in 2D. |
| Cell Intensity Std. Dev. (3D) | Standard deviation of voxel intensities in 3D. |
| NK Cell Score | Quantitative score indicating NK cell features. |
| Tumor Cell Score | Quantitative score indicating tumor cell features. |

Table A3: Synthetic parameters table.

| synthetic parameters | |
|---|---|
| num_components | 8 |
| interaction_radius | 10.0 |
| num_features_each_cell | 10 |
| distribution_low_comps | normal |
| sparsity_low_comps | False |
| T | 50 |
| n_distances_to_calculate | 100 |
| sparsity_percent | 0.2 |
| noise_low_rank_reco | 0 |
| noise_linear_reco | 0 |
| normalize_rows_columns | True |
| distribution_features | normal |
| update_of_features_if_no_effect | keep |
| fixed_A | False |
| scaling_features_method | standard |
| n_cell_type_rows | 2 |
| sigma_diff_graph | 0.3 |
| n_cells | 35 |
| building_w_thres | 69 |
| closest_As_to_consider | 10 |
| decor_components | True |
| include_plotting | True |
| n_unique_cells | 2 |
| with_timescales | False |
| with_gradient | False |
| interactions_style | tumor_nk |
| correlated_features_control | False |
| include_self_effect | True |
| distance_thres | 0.168793 |

# H  BASELINE IMPLEMENTATION DETAILS

## H.1  LINEAR DYNAMICAL SYSTEMS (LDS)

We fit a single, time-fixed feature-to-feature map $\boldsymbol{A} \in \mathbb{R}^{F \times F}$ by least squares over *all* ordered source–target cell pairs and time points. The model predicts target features at the next frame from source features at the current frame:

$$\widehat{\boldsymbol{A}} \;=\; \arg\min_{\boldsymbol{A}} \sum_{(s,n,t):\, s \neq n} \left\| \boldsymbol{x}_{t+1}^{(n)} \;-\; \boldsymbol{A}\,\boldsymbol{x}_t^{(s)} \right\|_2^2 ,$$

with no intercept term and no spatial/radius filtering. Let $\mathcal{I} = \{(s_i, n_i, t_i)\}_{i=1}^M$ index sampled ordered pairs and times, and define

$$\boldsymbol{X} \;=\; \begin{bmatrix} \boldsymbol{x}_{t_1}^{(s_1)} & \boldsymbol{x}_{t_2}^{(s_2)} & \cdots & \boldsymbol{x}_{t_M}^{(s_M)} \end{bmatrix} \in \mathbb{R}^{F \times M}, \qquad \boldsymbol{Y} \;=\; \begin{bmatrix} \boldsymbol{x}_{t_1+1}^{(n_1)} & \boldsymbol{x}_{t_2+1}^{(n_2)} & \cdots & \boldsymbol{x}_{t_M+1}^{(n_M)} \end{bmatrix} \in \mathbb{R}^{F \times M}.$$

The least-squares estimate solves $\min_{\boldsymbol{A}} \|\boldsymbol{Y} - \boldsymbol{A}\boldsymbol{X}\|_F^2$ with closed-form

$$\boldsymbol{A} \;=\; \boldsymbol{Y}\,\boldsymbol{X}^{\dagger},$$

where $\boldsymbol{X}^{\dagger}$ is the Moore–Penrose pseudoinverse (Penrose, 1956), estimated by 'sklearn.linalg' package. $\boldsymbol{X}^{\dagger} = \boldsymbol{X}^{\top}(\boldsymbol{X}\boldsymbol{X}^{\top})^{-1}$, hence $\boldsymbol{A} = \boldsymbol{Y}\,\boldsymbol{X}^{\top}(\boldsymbol{X}\boldsymbol{X}^{\top})^{-1}$. This baseline yields a single, global, time-fixed map shared across all pairs and frames. We included two versions of LDSs:

- Applied across all cells together (such that all cells can interact with all cells).
- Applied per cell, to enable higher expressivity, while limiting cross-cell interactions.

## H.2  SWITCHING LINEAR DYNAMICAL SYSTEM (SLDS)

In SLDS, the system can be in one of $K$ discrete states (modes), and in each mode, the dynamics follow a linear state space model. Particularly, at each time step $t$:

1. **Discrete state (switch variable):**
   - $z_t \in \{1, 2, \ldots, K\}$ - which dynamical mode we are in
   - Evolves as a Markov chain: $P(z_t | z_{t-1})$

   *In LICCHIE however, multiple processes can be co-active simultaneously through the decomposition $\boldsymbol{A}_t^{(n,k)} = \sum_{j=1}^J W_{j,t}^{(n,k)} \boldsymbol{c}_j \boldsymbol{r}_j^{\top}$.*

2. **Continuous latent state:**
   - $\boldsymbol{x}_t \in \mathbb{R}^d$ - hidden continuous state
   - In SLDS, dynamics depend on current mode $z_t$:

   $$\boldsymbol{x}_t = \boldsymbol{A}_{z_t} \boldsymbol{x}_{t-1} + \boldsymbol{B}_{z_t} \boldsymbol{u}_t + \boldsymbol{w}_t \tag{8}$$

   where $\boldsymbol{w}_t \sim \mathcal{N}(\boldsymbol{0}, \boldsymbol{Q}_{z_t})$.

   *In contrast LICCHIE captures explicit pairwise interactions: $\boldsymbol{x}_t^{(n)} = \sum_{k=1}^N \mathbb{1}_{d_{n,k}(t) \leq R} \boldsymbol{A}_t^{(n,k)} \boldsymbol{x}_{t-1}^{(k)}$, with spatial constraints and cell-specific matrices.*

3. **Observations:**

   $$\boldsymbol{y}_t = \boldsymbol{C}_{z_t} \boldsymbol{x}_t + \boldsymbol{D}_{z_t} \boldsymbol{u}_t + \boldsymbol{v}_t \tag{9}$$

   where $\boldsymbol{v}_t \sim \mathcal{N}(\boldsymbol{0}, \boldsymbol{R}_{z_t})$

- $\boldsymbol{A}_{z_t}$: Transition matrix for mode $z_t$ in SLDS
- $\boldsymbol{B}_{z_t}$: Input matrix
- $\boldsymbol{C}_{z_t}$: Emission/observation matrix
- $\boldsymbol{Q}_{z_t}, \boldsymbol{R}_{z_t}$: Process and observation noise covariances
- $\boldsymbol{u}_t$: Optional control input

*In contrast, LICCHIE's rank-1 underlying vectors $c_j, r_j$ represent interpretable source and target effects respectively.*

In SLDS, the discrete state follows a Markov chain with transition matrix:

$$P(z_t = j | z_{t-1} = i) = \pi_{ij} \tag{10}$$

This allows the system to switch between different linear dynamics.

*In contrast, LICCHIE's interactions evolve smoothly through distance-weighted regularization $\|A_t^{(n,k)} - A_{t'}^{(n',k')}\|_2^2 \leq \epsilon(\delta)$, avoiding discrete jumps.*

At last, when SLDS is applied to feature space, it learns feature-to-feature transitions but cannot distinguish within-cell evolution from cross-cell interactions without scaling to dimension $\mathbb{R}^{Nm \times Nm}$.

We implemented Switching Linear Dynamical System (SLDS) baselines using the ssm package (Linderman et al., 2020). The SLDS model was configured with $K = 10$ discrete latent states to match LICCHIE's number of components ($J = 10$), and the latent dimensionality $D$ was set equal to the number of observed features. We used Gaussian identity emissions (emissions='gaussian_id') to directly model the observed feature space without additional transformations. The model was fit using black-box variational inference (BBVI) with a mean-field variational posterior, running for 500 iterations.

We fitted an independent SLDS for each cell trajectory (interactions are implicit within that cell's feature space; Fig. S3)

Notably, since SLDS does not produce components that can co-occur at the same time point, it cannot separate the evolution of multiple driving forces acting together, nor does it account for unit-to-unit (e.g., cell-to-cell) distance. In addition, since we applied SLDS in feature space—where each latent transition represents feature-to-feature effects (i.e., nodes are features, not cells)—it does not naturally distinguish interactions within the same cell from interactions across different cells. Extending SLDS to capture these aspects would require increasing the number of nodes to $n \times k$, ($n$ the number of cells and $k$ the number of features), which would be extremely large and thus hinders interpretability.

### H.3    GRAPH NEURAL NETWORKS (GNNs)

To contextualize LICCHIE within the broader landscape of methods for spatiotemporal cellular data, we compared against a Graph Neural Network (GNN) baseline implemented in PyTorch Geometric (Fey et al., 2025). Each frame $t$ is represented as a graph whose nodes are cells and whose edges connect spatial neighbors with distance-weighted strengths ($w_{n,k}(t) = \exp(-d_{n,k}(t)/R)$, with $R$ equivalent to the distance used for LICCHIE). A message-passing network is trained to predict the feature set $x_t$ from $x_{t-1}$ by minimizing mean-squared error using Adam.

We construct a GNN with 3 graph convolutional layers, 64 hidden channels, ReLU activations, batch normalization, and dropout ($p = 0.1$). Training was performed with Adam optimizer at learning rate 0.001, batch size 16, and 100 epochs with early stopping (patience 20). Training was performed on GPU using PyTorch 2.0+ and PyTorch Geometric 2.3+.

Table A4 summarizes the key differences between the GNN approach and LICCHIE. Broadly, GNNs learn implicit interactions through message passing in a black-box neural network– they achieve high expressivity through non-linear transformations but require post-hoc explanation methods to interpret learned representations (Fig. S4). In contrast, LICCHIE explicitly recovers interpretable interaction matrices $A_t^{(n,k)}$ where each entry $[A_t^{(n,k)}]_{i,j}$ quantifies the effect of source feature $j$ and cell $k$ on target feature $i$ of cell $n$. Further, LICCHIE's low-rank decomposition $A_t^{(n,k)} = \sum_{j=1}^{J} W_{j,t}^{(n,k)} c_j r_j^\top$ reveals modular interaction components that can be individually interpreted and manipulated, enabling mechanistic hypothesis testing through targeted weight perturbations—capabilities absent in standard GNN architectures.

### H.4    EVALUATIONS

- **Quantitative:** accuracy of the inferred "interaction matrices" compared to ground truth construction. The "interaction matrices" per method are defined as:

Table A4: Comparison of GNN and `LICCHIE` approaches for learning cell-cell interactions.

| Aspect | GNN | `LICCHIE` |
|---|---|---|
| Interactions | Implicit via message passing | Explicit pairwise matrices $\boldsymbol{A}_t^{(n,k)}$ |
| Interpretability | Post-hoc explanation needed | Direct feature-to-feature effects |
| Structure | Black-box neural network | Low-rank with interpretable components |
| Discovery | Distributed representations | Explicit modular components $\{c_j, r_j\}$ |
| Intervention | Hard to target mechanisms | Modulate component weights $W_{j,t}^{(n,k)}$ |

1. Linear: a single feature-to-feature matrix; $\boldsymbol{A}$ for $\boldsymbol{x}_t \approx \boldsymbol{A}x_{t-1}$.
2. Per-cell Linear: per-cell feature-to-feature matrices; $\boldsymbol{A}^{(n)}$ for $\boldsymbol{x}_t^{(n)} \approx \boldsymbol{A}^{(n)}\boldsymbol{x}_{t-1}^{(n)}$.
3. Per-cell SLDS: per-cell transition matrices; $\boldsymbol{A}_t^{(n)}$ for $\boldsymbol{x}_t^{(n)} \approx \boldsymbol{A}_{z|t-1}^{(n)}\boldsymbol{x}_{t-1}^{(n)}$.
4. GNN: approximated transition matrices, propagating the GNN loading matrices from the feature space of one time point to the next via the feature vector.
5. `LICCHIE`: the learned interaction matrices.

- **Qualitative:** visualization of the methods' outputs and comparison to expected ground-truth dynamical operators.

## I METHODS PERFORMANCE COMPARISON ON REAL-WORLD APPLICATION

In order to compare `LICCHIE` to other methods in a real-world scenario, we conducted a series of evaluations to test each model's ability to predict biologically meaningful properties of the tumor-NK interaction system. Namely, we rely on known properties of cells in the data—the cell type (tumor, NK) and changes in their viability status (tumor) or cell polarization (NK), and test the ability of the models' outputs to accurately predict them.

First, we define a consistent set of seven properties describing the interactions identified by each method. We used these properties (hereafter 'output features') to predict the involved cell types and their polarization or viability status over time. These output features include: (i) total influence (sum of all elements), (ii) maximum column sum (strongest influence on any target feature), (iii) maximum row sum (strongest influence from any source feature), (iv) diagonal sum (self-feature interactions), (v) Frobenius norm (overall matrix magnitude), and (vi, vii) maximum/minimum elements (extreme interaction strengths). These features were used as input to a logistic regression classifier with balanced class weights to test each method's ability to capture distinct cell and interaction properties.

To address class imbalance in the dataset, we employed two strategies: (1) balanced class weights in logistic regression, and (2) sub-sampling approaches where we resampled both classes to equal sizes using bootstrap resampling with replacement.

**Source cell type Prediction** For each interaction at time $t$, we extracted the above output features to predict the identity of the source cell (NK vs. tumor) via logistic regression with cross validation. Notably, while the baselines accuracy was close to random, GNN (0.54) and SLDS (0.57), `LICCHIE` presented higher predictive power (0.71) and was found to be less sensitive to data imbalance (Fig. S8a). Furthermore, looking at the confusion matrices, it is apparent that baseline methods show severe prediction collapse with degenerate concentration in single classes. `LICCHIE` on the other hand, produces balanced predictions for both cell types, maintaining strong performance even with balanced sampling (`LICCHIE`: 0.7, GNN: 0.5, SLDS: 0.5; Fig. S8b).

**Target cell type prediction** Similarly to the previous setting we now predict the identity of the target cell. `LICCHIE` again demonstrates better performance (0.68) compared to baseline methods (GNN: 0.54, SLDS: 0.57; Fig. S8c), maintaining meaningful predictions across both classes while baselines exhibit prediction bias with near-zero off-diagonal elements. With balanced sampling, `LICCHIE` achieves accuracy of 0.71 compared to baselines depicting random-like results (GNN: 0.51, SLDS: 0.5).

**Interaction type classification**   We extended the binary classification to predict the specific source→target cell type combination, yielding four classes (NK→NK, NK→Tumor, Tumor→NK, and Tumor→Tumor). This test aims to test the models' ability to capture different interaction patterns dictated by identities of interacting cells. While baselines remain at close to random performance, `LICCHIE` maintains good performance (`LICCHIE`: 0.60, GNN: 0.26, SLDS: 0.32; Fig. S8a).

**Cellular state transition prediction**   At last, we turn into a temporally resolved test–checking the ability to predict cellular states at both current time $t$ and future time $t + 1$. For each interaction matrix $A_t^{(n,k)}$ extracted by each method, we predicted the status of target cell $n$ both at time $t$ and at time $t + 1$. We conducted separate analyses for NK cells (predicting polarization status: non-polarized vs. polarized) and tumor cells (predicting viability: alive vs. dead). Only samples with non-NaN states were included in the analysis. `LICCHIE` shows better performance on all prediction tasks (Fig. S8a,d), achieving above-chance accuracy for the challenging NK polarization prediction ($t$: 0.62, $t + 1$: 0.65) while baselines remain at random-like performance (GNN $t$: 0.23, $t + 1$: 0.22; SLDS $t$: 0.23, $t + 1$: 0.23).

To summarize, across all evaluation tasks `LICCHIE` maintains consistently high accuracy ($> 0.6$) while baselines tend to follow random-like accuracy (Fig. S8a). The lower performance of the baseline methods suggests that their learned interactions cannot fully capture these existing biological components in the real-world data (or at least that a more elaborate post-processing is required to extract the information from their outputs). This is in line with the synthetic evaluations performed, and can be explained by considering the nature of these baselines; SLDS, while interpretable, produces transitions with abrupt switches between them (Fig. S9) that cannot capture the smoothly time- and distance-varying nature of cell-to-cell interactions; GNN produces loading matrices that require extra processing to recover approximate transitions, and these approximations change over time in ways that are hard to interpret because they lack an underlying interaction basis (Fig. S10).

# J SUPPLEMENTARY FIGURES

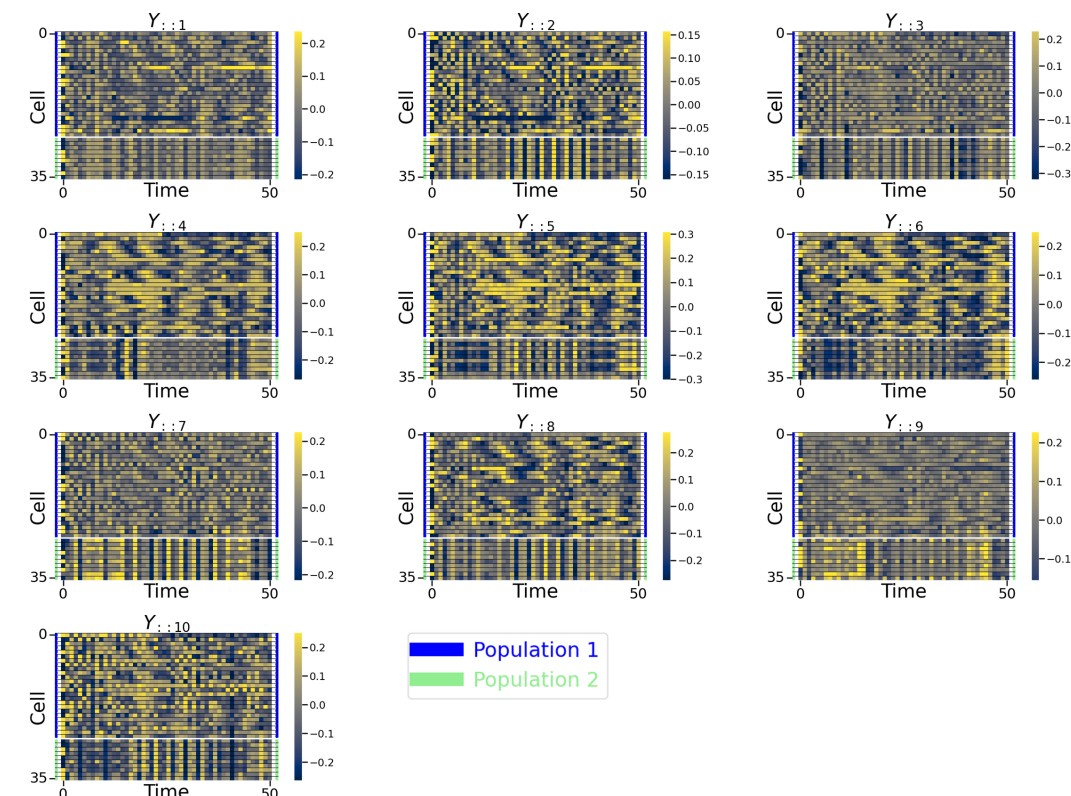

Figure S1: *Overview of the generated synthetic data.* Each subplot presents values of a given feature over all cells across time.

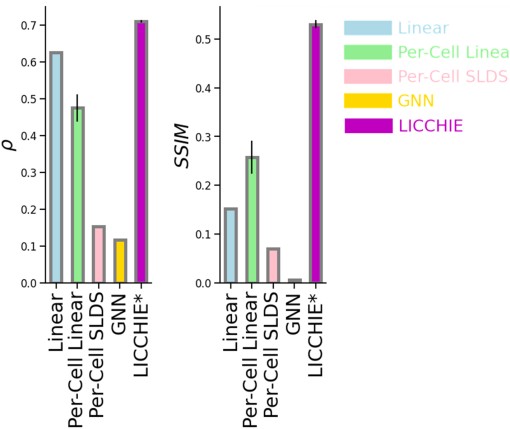

Figure S2: *Additional evaluations over synthetic data.* A comparison of `LICCHIE` to all baselines (linear dynamics, per-cell linear approximation, per-cell SLDS, and GNN) considering (left) correlation of identified interactions with the ground truth , and (right) SSIM score (Nilsson & Akenine-Möller, 2020)

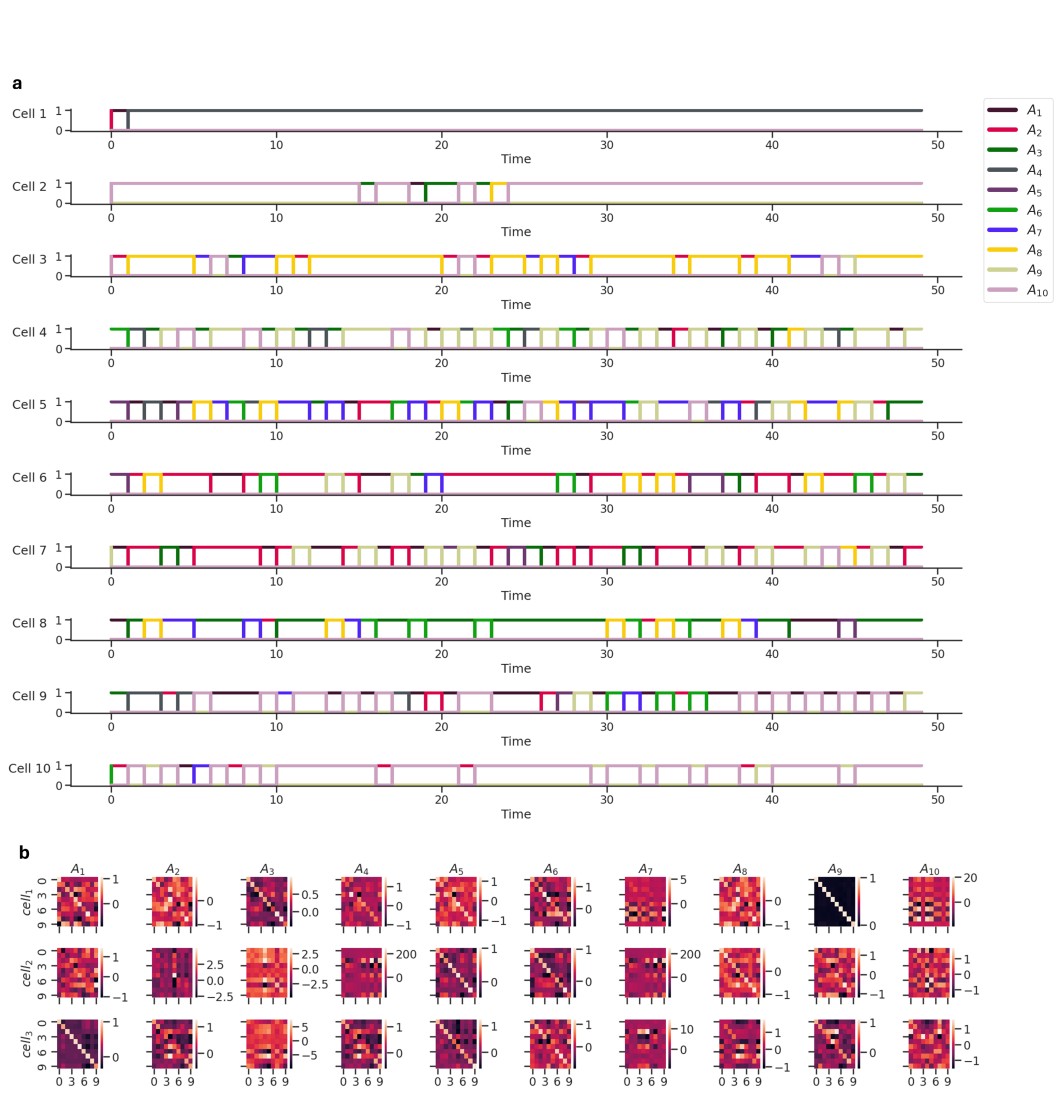

Figure S3: *Per-cell SLDS baseline components on synthetic data.* **a**, Extracted weights $\boldsymbol{W}_t^{(k)}$ for the respective transition matrix, $\boldsymbol{A}^{(n,k)}$ with $k \in [1, 10]$ and for $n \in [1, 10]$ (a subset of the total 35 cells), presented over time, $t \in [0, 50]$. **b**, Example of the set of per-cell SLDS transition matrices, $\boldsymbol{A}^{(n,k)}$ with $k \in [1, 10]$, and for $n \in [1, 10]$ (a subset of the total 35 cells) identified by SLDS.

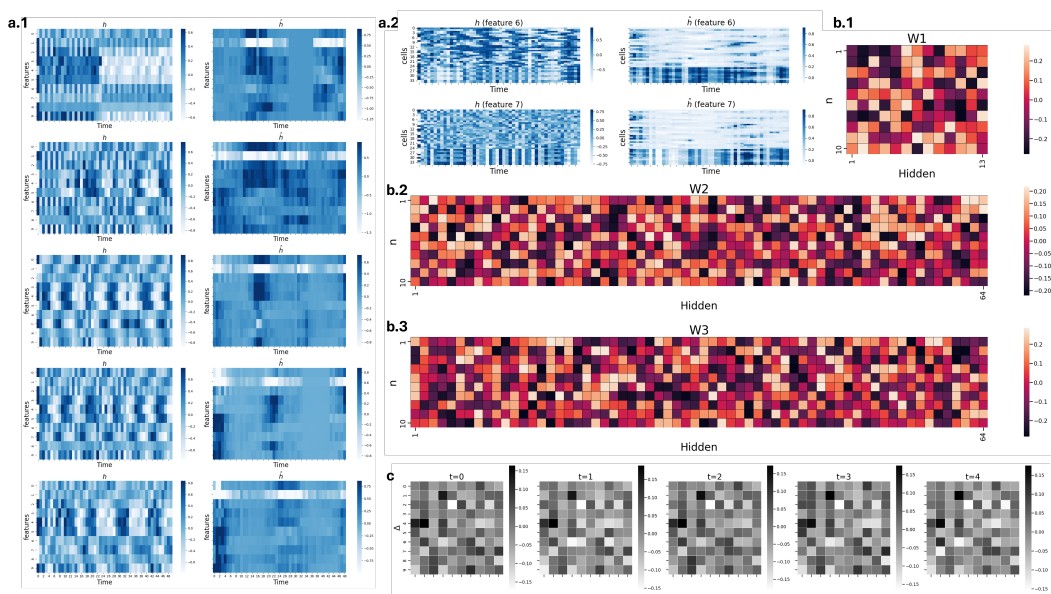

Figure S4: *GNN-based modeling of the synthetic data.* **a**, Feature space reconstruction evaluation. **a.1** Per-cell reconstruction comparison across different cells (left: ground truth, right: reconstructed). **a.2** Per-feature reconstruction for two example features, comparing ground truth ($h$) with predictions ($\hat{h}$). **b**, The loading (weight) matrices of the three layers in the trained GNN; **b.1**, **b.2**, **b.3** respectively. **c**, Approximated transition matrices for first 5 time points, reconstructed using the loading matrices in subplot (**b**) and the feature values at each time point.

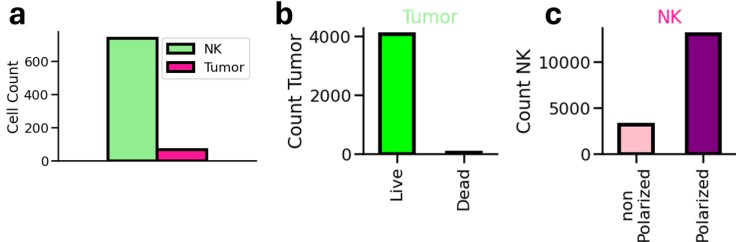

Figure S5: *Tumor-NK data overview.* **a**, Total number of observations for tumor cells (across all time points). **b**, Total number of observations for NK cells (across all time points). **c**, Number of unique NK vs tumor cells.

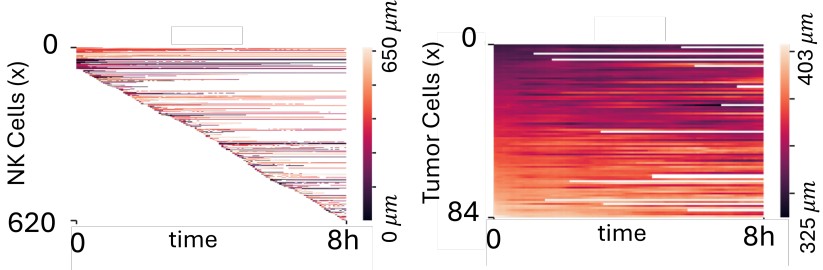

Figure S6: *Transient cell activations observed in the data*

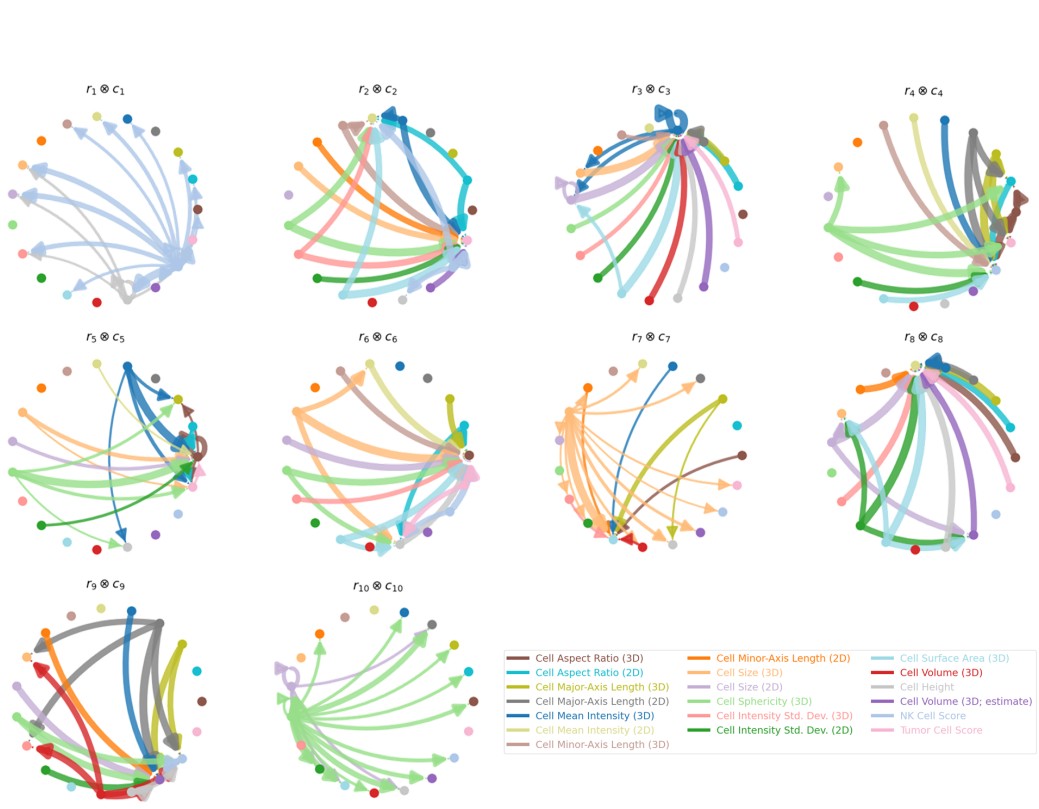

Figure S7: *Identified components by* `LICCHIE` *reveal variability in interaction structures*. Structures include *localized targets* (e.g., $M_2$, $M_4$), *source-emerging* (e.g., $M_1$, $M_{10}$), self *feedback-like* interactions (e.g., $M_3$) and multi-step interactions, including direct and indirect influences (e.g., $M_6$).

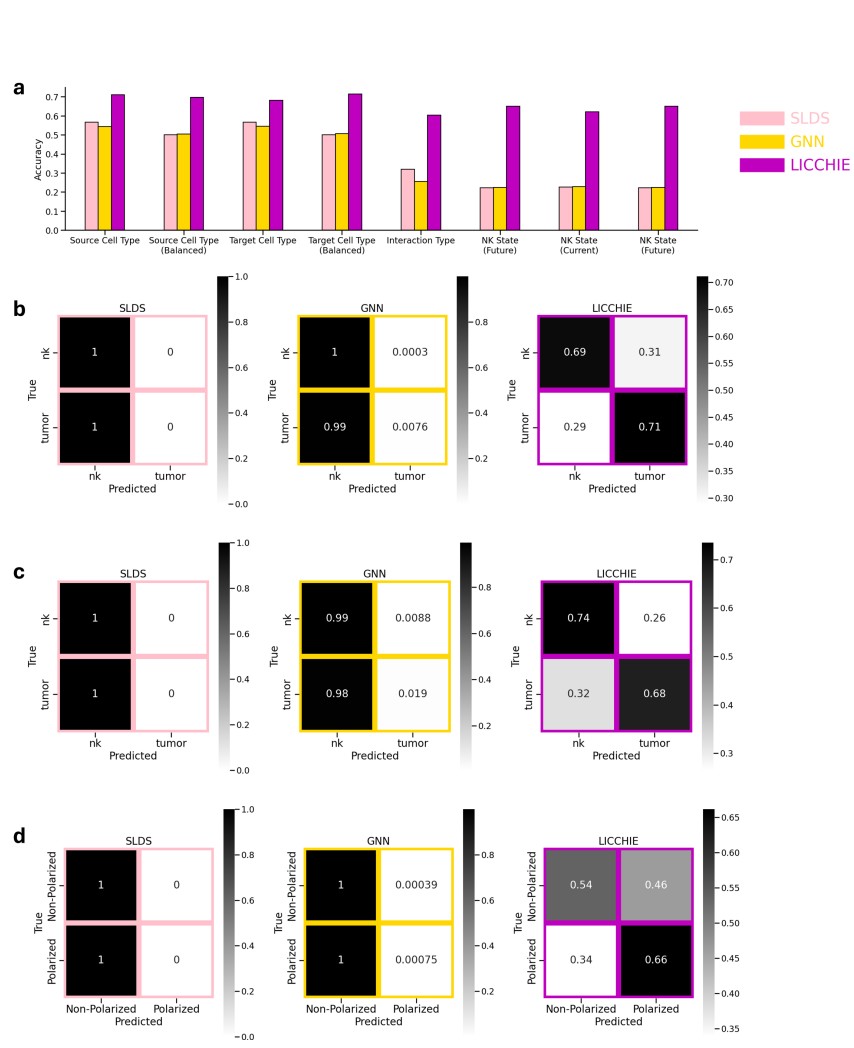

Figure S8: *Performance comparison of* LICCHIE *against baseline methods (SLDS, GNN) on real-world data using machine learning evaluation tasks.* **a**, Classification accuracy across different prediction tasks. For each task, we trained logistic regression classifiers with balanced class weights on interaction matrix features, comparing performance with and without sub-sampling (addressing class imbalance). **b-d**, Normalized confusion matrices showing prediction performance for: (**b**) source cell type classification, (**c**) target cell type classification, and (**d**) NK cell polarization state prediction. Each method's interaction matrices were used to extract a set of features consisting of seven statistical features (sum, max column/row sums, diagonal sum, Frobenius norm, max/min elements) as input to the classifiers.

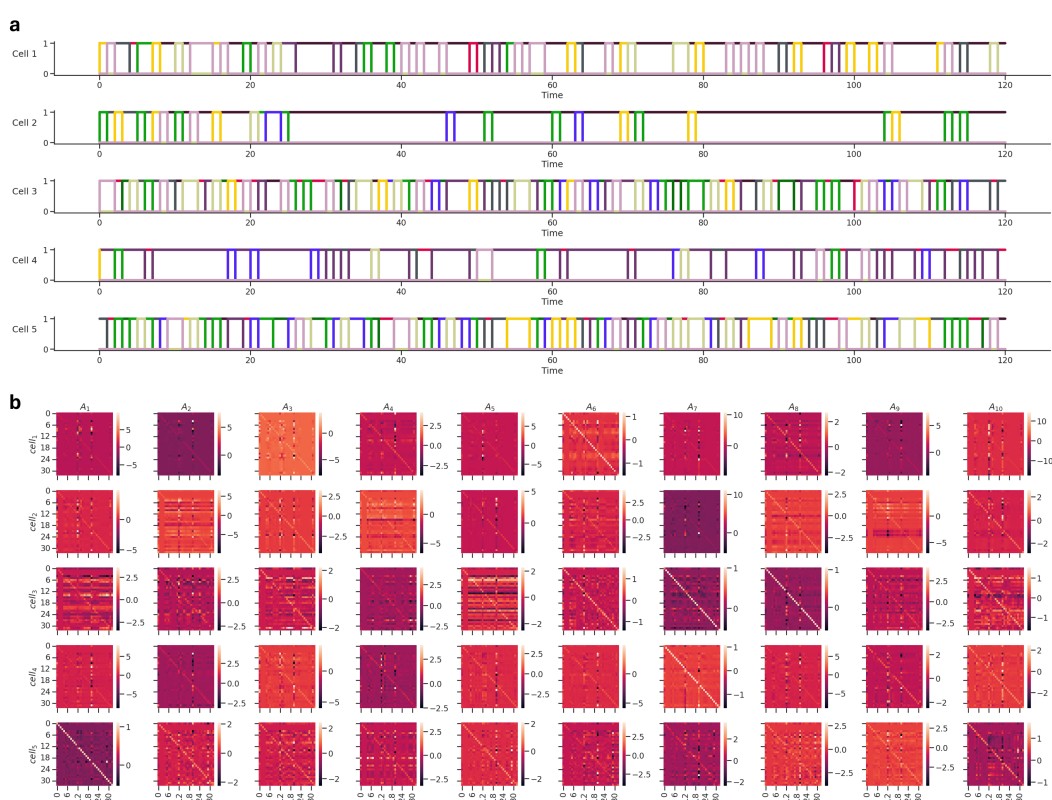

Figure S9: *Per-cell SLDS real-world data components*. **a**, Switched activation patterns of transition matrices for first 5 cells. **b**, Per-cell transition matrices for first 5 cells.

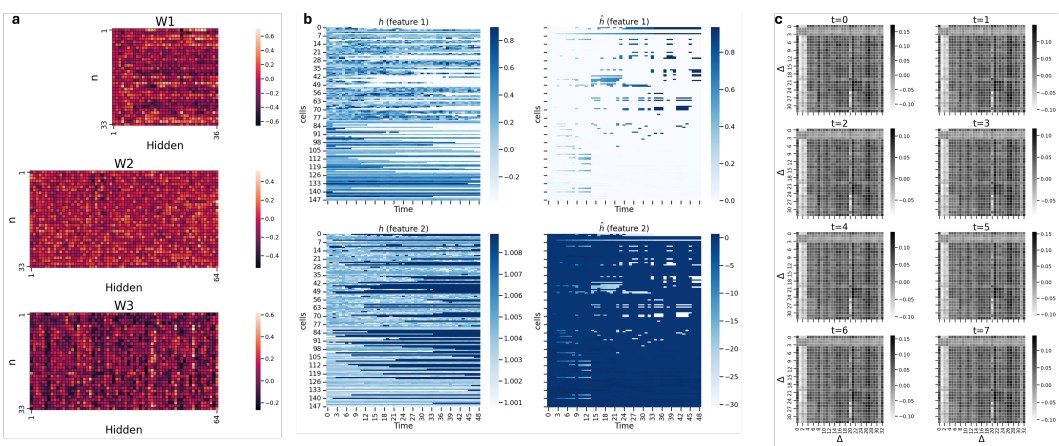

Figure S10: *Components identified by the GNN baseline on Real World Data*. **a**, Loading matrices from GNN trained on the real-world NK-tumor data. **b**, Example reconstruction for first two features, $h$ ground truth, $\widehat{h}$ prediction results. **c**, Approximated transition matrices from loadings and latent states for the first 8 time points.

