# OpenReview forum: "Low-rank Interpretable Cell–Cell Hidden Interactions from Embeddings"
_ICLR.cc/2026/Conference — Submitted to ICLR 2026_

### Official Review · Reviewer_Siuv · 2025-10-30

**Soundness:** 2
**Presentation:** 3
**Contribution:** 2
**Rating:** 2
**Confidence:** 3

**Summary:**

This paper introduces LICCHIE — a low-rank dynamic model designed to infer time-varying cell–cell interactions” from live-cell imaging data. Each cell is represented as a temporal feature vector derived from morphological and spatial descriptors, and pairwise influences between cells are modeled as time-dependent linear transformations. These interactions are constrained to be low-rank, sparse, and temporally smooth, aiming to balance interpretability and expressiveness. The authors validate LICCHIE on synthetic datasets and apply it to live-cell microscopy of tumor–NK co-cultures, claiming that the learned interaction matrices reveal biologically meaningful influence patterns and potential regulatory mechanisms.

**Strengths:**

**Readable presentation:**

The manuscript is well-written and clearly organized, with intuitive figures that make the mathematical setup easy to follow even for non-domain experts.

**Mathematical clarity:**

 The formulation is compact and well-structured, combining ideas from linear dynamical systems, low-rank decomposition, and temporal smoothness regularization in an interpretable way.

**Weaknesses:**

**Conceptual and biological validity concern**

The paper defines cell–cell interactions from live-cell imaging sequences, yet the experimental setup appears to involve isolated cells or loosely adherent co-cultures rather than structured tissue or organoid environments.  In such conditions, cells are not embedded within a continuous microenvironment, and long-range or contact-based signaling is largely absent.  Therefore, what the model captures is more accurately morphological co-variation or motion correlation among nearby cells, rather than bona-fide cell–cell communication.  This distinction should be made explicit, as it fundamentally affects the biological interpretation of the inferred interaction matrices.

**Lack of molecular or mechanistic validation**

 The inferred interaction matrices are not validated against known ligand–receptor signaling pathways, transcriptional profiles, or perturbation responses.  Without molecular or experimental corroboration, it remains unclear whether the model captures meaningful communication or merely statistical dependencies.  This weakens the biological interpretability of the proposed “interaction components.

**Questions:**

None

---

> ### Author Response · Authors · 2025-11-24
>
> Dear Reviewer,
>
> Many thanks for your comments on the clarity of the mathematical formulation of LICCHIE (**"The formulation is compact and well-structured"**), the papers’ organization and visual presentation (**"…clearly organized, with intuitive figures"**) and our writing (**“The manuscript is well-written"**).
>
>
> Below we address the two main concerns and describe specific revisions (all marked in **blue** in the revised manuscript submission):
>
>
>
> **Conceptual and biological validity concern**
>
> **Concern:** _The model may capture morphological co-variation rather than bona-fide cell–cell communication in the loosely adherent co-cultures._
>
> **Clarification**: We agree with the reviewer that cell-cell interactions shall be analyzed with respect to the experimental setting, so while the generality of LICCHIE makes it applicable to a broad range of systems, the analysis and interpretation must take into account the exact setting. In the presented work, as a real data application of LICCHIE, we analyzed 3D live-cell imaging of patient-derived tumor organoids co-cultured with primary human NK cells (Liu et al. [1]; **Section 5.2**). Such experimental systems are widely used to interrogate cell–cell interactions in a controlled 3D microenvironment and are well suited for motile immune–tumor co-cultures in which contact-dependent interactions are frequent and mechanistically consequential—even in the absence of long-range paracrine fields. Multiple live-imaging studies have established that brief, often serial engagement events are the dominant mode of cytotoxic action, not hours-long stable conjugates. For CD8⁺ and engineered T cells in organoid settings, variation in contact within minutes-scale windows and serial engagement strongly predict killing, and single CD8⁺ effectors can eliminate many targets in succession [2, 3]. Complementary NK-cell work demonstrates a similar phenomenon and quantifies burst-like killing after short contacts [4, 5]. These observations motivate our modeling focus on time-varying, local influences between nearby cells, and guide our interpretation of the NK–tumor interaction motifs recovered by LICCHIE. Importantly, similar co-culture imaging readouts have proven translational value, capturing predictive information about effector potency and target susceptibility, with demonstrated utility for assessing engineered T-cell products in patient-derived organoids [2, 3, 6]. By learning distinct, time-localized modes of interaction from such data, LICCHIE offers a principled and interpretable framework for dissecting immune–tumor influence patterns that matter for therapeutic design.
>
> **Revision**: We would like to thank the reviewer for this comment as it highlighted two overlooked aspects in the manuscript which we have now revised. First, we added the above contextualization to the revised manuscript (**Section 5.2.**), clarifying the biological relevance of the provided interpretations for the real-data application. Second, we emphasize the need to analyze the inferred interactions with respect to the experimental design, as the interactions crucially depend on the multicellular environment (**Section 6**).
>
> 1. Liu et al. Deep Learning–Based 3D Single-Cell Imaging Analysis Pipeline Enables Quantification of Cell–Cell Interaction Dynamics in the Tumor Microenvironment. Cancer Research 84.4: 517-526 (2024).
>
> 2. Dekkers, J. F. et al. Uncovering the mode of action of engineered T cells in patient cancer organoids. Nature Biotechnology 41, 60–69 (2023). https://doi.org/10.1038/s41587-022-01397-w
>
> 3. Alieva, M. et al. BEHAV3D: a 3D live imaging platform for comprehensive analysis of engineered T-cell behavior and tumor response. Nature Protocols (2024). https://doi.org/10.1038/s41596-024-00972-6
>
> 4. Choi, P. J. & Mitchison, T. J. Imaging burst kinetics and spatial coordination during serial killing by single natural killer cells. PNAS 110(16):6488–6493 (2013). https://doi.org/10.1073/pnas.1221312110
>
> 5. Vanherberghen, B. et al. Classification of human natural killer cells based on migration behavior and cytotoxic response. Blood 121(8):1326–1334 (2013). https://doi.org/10.1182/blood-2012-06-439851
>
> 6. Logun, I. et al. Patient-derived glioblastoma organoids as real-time avatars for assessing responses to clinical CAR-T cell therapy. Cell Stem Cell (2024). https://doi.org/10.1016/j.stem.2024.11.010

---

> > ### Author Response · Authors · 2025-11-24
> >
> > *Lack of molecular or mechanistic validation**
> >
> > **Concern:** _No validation against ligand–receptor, transcriptional profiles, or perturbations._
> >
> > **Clarification:** The Liu et al. [1] assay provides phenotypic ground truth visible in the time-series acquisition. In our analysis we validate LICCHIE through its inference power–namely, the learned weights $W$, allow accurate prediction of the cellular phenotypes on held-out frames, and controlled weight “sweeps” modulate predicted NK polarization and tumor-death scores in plausible, component-specific ways (**Fig. 4**). These results show that learned interaction motifs carry phenotype-relevant signals without using phenotype labels during training–providing a phenotypic validation.
> >
> > We agree that additional experimental measurements of molecular features and the effects of perturbations on the interaction motifs identified by LICCHIE would be valuable; however they are outside the scope of the current manuscript. In addition, it is important to note that, current Ligand-Receptor inference frameworks from scRNA-seq are themselves proxies, limited by database incompleteness, expression–protein discordance, lack of kinetics, and challenges in benchmarking without ground truth [7-11]. Recent quantitative reviews and method evaluations underscore these limitations and call for orthogonal, dynamic readouts to assess functional influence [7-10]. Our contribution is to provide exactly such a readout: continuous experimental measurements of cell-state change under naturalistic contact dynamics.
> >
> > **Revision:** In the revised manuscript we discuss the suggested limitation of the current work and strongly advocate for experimental validation with future usages of LICCHIE; we added language noting that molecular mechanism mapping is future work and that LICCHIE provides a scaffold upon which multi-omic integration (imaging ↔ scRNA-seq/spatial) can be layered in subsequent studies (**Section 6**).
> >
> >
> > 7. Liu, Y., Beyer, A. & Aebersold, R. On the Dependency of Cellular Protein Levels on mRNA Abundance. Cell 165, 535–550 (2016). https://doi.org/10.1016/j.cell.2016.03.014
> >
> > 8. Armingol, E., Officer, A., Harismendy, O. & Lewis, N. E. Deciphering cell–cell interactions and communication from gene expression. Nat. Rev. Genet. 22, 71–88 (2021). https://doi.org/10.1038/s41576-020-00292-x
> >
> > 9. Armingol, E., Baghdassarian, H. M. & Lewis, N. E. The diversification of methods for studying cell–cell interactions and communication. Nat. Rev. Genet. 25, 381–400 (2024). https://doi.org/10.1038/s41576-023-00685-8
> >
> > 10. Dimitrov, D. et al. Comparison of methods and resources for cell–cell communication inference from single-cell RNA-Seq data. Nat. Commun. 13, 3224 (2022). https://doi.org/10.1038/s41467-022-30755-0
> >
> > 11. Cesaro, G., Nagai, J. S., Gnoato, N., Chiodi, A., Tussardi, G., Klöker, V., Musumarra, C. V., Mosca, E., Costa, I. G., Di Camillo, B., Calura, E. & Baruzzo, G. Advances and challenges in cell–cell communication inference: a comprehensive review of tools, resources, and future directions. Briefings in Bioinformatics 26(3): bbaf280 (2025). https://doi.org/10.1093/bib/bbaf280
> >
> >
> > ​

---

### Official Review · Reviewer_j8Lv · 2025-10-31

**Soundness:** 4
**Presentation:** 4
**Contribution:** 4
**Rating:** 8
**Confidence:** 4

**Summary:**

The paper presents LICCHIE, a low-rank, interpretable framework for modelling cell-cell interactions from live-cell imaging data.  It represents pairwise influences between cells as combinations of shared rank-1 “interaction motifs,” enabling biologically interpretable analysis of how cellular features dynamically affect one another over time.

**Strengths:**

- The problem is very well articulated and follows a very coherent flow.
- The approach is biologically grounded and is well-motivated at every architectural component.

**Weaknesses:**

- In the abstract, NK is not defined.
- Ambiguity in biological grounding and temporal consistency: The decision to avoid explicit cell tracking and instead operate purely in “feature space” is only loosely justified in Section 3. Without clear temporal correspondence between specific cells across frames, it is uncertain how the model distinguishes genuine dynamic interactions from coincidental correlations among transient feature observations.
- Lack of clarity in feature-space formulation and identity handling: While the authors justify avoiding explicit cell tracking by operating in “feature space,” it remains unclear how temporal consistency is maintained when cell identities are not preserved. Without explicit linking across frames, it is ambiguous how the model distinguishes genuine temporal evolution of a single cell from population-level variability or measurement noise.

**Questions:**

None

---

> ### Author Response · Authors · 2025-11-24
>
> Dear Reviewer,
>
>
> We are very grateful for your positive review, particularly the fact that you found that the problem was well presented and that the suggested approach was well-motivated and biologically set.
>
> Let us answer some of the issues you have raised (revisions are marked in **blue** in the new manuscript submission).
>
> **Weaknesses**:
>
> [1] **NK definition**
>
> **Concern:** _In the abstract, NK is not defined._
>
> **Revision:** Thank you for raising, we added the definition, Natural Killer cells (NK) in the revised version.
>
> [2] **Ambiguity in biological grounding and temporal consistency**
>
> **Concern:** _The decision to avoid explicit cell tracking and instead operate purely in “feature space” is only loosely justified in Section 3._
>
> **Clarification:** We thank the reviewer for the opportunity to clarify this point here, and follow with modifications in the revised manuscript. Importantly, LICCHIE does not discard temporal correspondence; by design it enforces short-term tracking. Each interaction matrix $A^{(n,k)}_t$ captures the evolution of the same cell $n$ from time point $t-1$ to $t$ via its neighboring cells (**Fig. 1, Section 4; Eq. 1**). This preserves a concrete temporal ordering and physical neighborhood for every interaction term while avoiding dependence on long-range tracking, which is a challenging task  and prone to errors [1]. In short, LICCHIE retains short-horizon identity where it is reliable and performs inference in feature space to aggregate across contexts—thereby isolating cell–cell interactions without propagating uncertain identities over longer trajectories.
>
> Beyond this, we believe LICCHIE’s design choices prevent learning random correlations among transient feature observations:
>
> 1. *Low-rank interaction-features regularization*: the core sparsity regularizer which restricts the motif space (**Section 4; Eq. 3**). Notably, it is less likely that non-informative correlations will be consistently aligned with the same limited shared motifs across many pairs and times.
>
> 2. *Locality constraint*: fitting interactions over a restricted spatial neighborhood reduces spurious influence from distant cells (**Fig. 1, Sections 3, and 4**).
>
> 3. *Cross-interaction similarity*: this loss term enforces a constraint that interactions vary smoothly across pairs with similar stacked source–target features, requiring re-occurrence of learnt interaction motifs, avoiding learning “singular” events (**Section 4, Eq. 2, App. C**)
>
> **Revision:** We clarify this point in the revised text (**Fig. 1, Sections 1, 3, 4, and 6**).
>
> [3] **Feature-space formulation & handling of identity**
>
> **Concern:** _it remains unclear how temporal consistency is maintained when cell identities are not preserved._
>
> **Clarification:** To recap the above, we realize our description “Operating in feature space” was unclear; our intention was to convey that we avoid propagating cell-level identities across long sequences, but we certainly do not discard them, we rather force short-range association, $t-1$ to $t$. Specifically, we learn pairwise, time-specific interaction matrices under the locality, smoothness and sparsity constraints, as detailed in response **2** (**Section 4**). In addition, the model allows incorporating population specific features (**App. B**), separating population variability from single-cell temporal evolution. Now, since the model forces learning causal transitions $t-1$ to $t$ the temporal consistency is maintained. Following this, we believe the design choices described in **2** explicitly disentangle genuine temporal evolution from random “events”.
>
> **Revision:** We clarify this point in the revised text (**Fig. 1, Sections 1, 3, 4, and 6**).
>
> 1. Maška et al. The cell tracking challenge: 10 years of objective benchmarking. Nature Methods 20.7: 1010-1020 (2023). https://doi.org/10.1038/s41592-023-01879-y

---

### Official Review · Reviewer_J9Kt · 2025-11-01

**Soundness:** 3
**Presentation:** 3
**Contribution:** 4
**Rating:** 8
**Confidence:** 4

**Summary:**

This paper develops LICCHIE, a model for inferring cell-cell interactions from live imaging of biological cell data. The model learns time-varying interaction matrices that are constrained by spatial proximity and a low-rank regularization. Optimization is performed by iterative linear regression and canonical polyadic tensor decomposition (PARAFAC).

**Strengths:**

1.	The formulation of the biological problem is, to my knowledge, unique. I have not seen any papers tackling this problem previously. This is an important biological problem, however, and an initial approach to solving it is an important contribution.
2.	The model formulation is sensible and parsimonious, but also interesting from an ML perspective.
3.	The biological insights gained from running the model on real data are quite interesting. It seems that this method provides new important new insights into these kinds of live-cell imaging datasets.
4.	Evaluations using simulated data show that the method accurately recovers ground truth interactions compared to baseline models.

**Weaknesses:**

1.	Baselines used for simulated data comparison are a bit simplistic. I understand that there are not really competing methods to compare with here. But, it seems that you could for example ablate your model more to understand which parts of the objective are most important for strong performance.
2.	Comparisons with baselines on the real data are missing.

**Questions:**

1.	Using a tensor decomposition here is an interesting approach. You imposed a low-rank constraint to obtain rank-1 components. Does it make biological sense to consider a tensor decomposition with a core tensor instead (Tucker decomposition)? This could give a different perspective on the structure of the cell interactions.
2.	The paper mentions cell state several times, but this seems largely aspirational, unless I misunderstood the details of the real dataset. Can you clarify what “cell state” means in the real data application presented here? If you’re not modeling it here, what sort of data would you need to do true cell state modeling?

---

> ### Author Response · Authors · 2025-11-24
>
> Dear Reviewer,
>
> We felt encouraged by your overall positive assessment of our paper, notably on the novelty in formulation of the biological problem (“**to my knowledge, unique.**”) and its importance (“**approach to solving it is an important contribution.**”), as well as the interpretability of the model and the biological insights it provides (“**this method provides new important new insights**”).
>
> Let us relate to the weaknesses and questions you have raised (we marked the referred revisions in **blue** in the revised manuscript submission).
>
> **Weaknesses**:
>
> [1] **Baseline comparisons for simulated data**
>
> **Concern:** _Baselines used for simulated data comparison are a bit simplistic._
>
> **Clarification:** As the reviewer accurately noted, there are limited comparable methods that enable direct 1-to-1 component comparison. Namely, most approaches fail to provide direct interpretable dynamical operators analogous to the LICCHIE output. With this, to strengthen our evaluation, we have extended the comparisons to additional approaches from which we can estimate approximate components: Switching Linear Dynamical Systems (SLDS [1]) fit per-cell and Graph Neural Networks (GNN), set with distance-weighted spatial graph and trained to predict the features evolution at $x_{t}$ from $x_{t-1}$ (**Section 5.1, Fig. 2, App. H**).
>
> For both methods we provide quantitative and qualitative analysis. Quantitatively we measure the accuracy of the inferred “interaction matrices” of each method; For SLDS we use the learnt transition matrices and for GNN we obtain an approximation using the loading (weight) matrices and the feature values at each time point (details provided in **App. H**). We found that LICCHIE remains the best performing method, capturing most accurately the ground truth structure (**Fig. 2, Fig. S2**)
>
> Qualitatively we visualize the outputs of each method. We observe that while SLDS can capture changes over time, it simplifies the changes to switching instances, and is not designed to capture transient cell observations as often encountered in live cell imaging data. Also, unlike LICCHIE, it cannot separate the interaction evolution to its multiple co-present underlying processes (**App. H, Fig. S3**).
>
> The GNN baseline does not yield any component that can naturally be used to interpret the underlying interactions. With that we visualize the feature space reconstruction, demonstrating it fails to capture the true structure, the loading (weight) matrices, which demonstrate a non-sparse interaction pattern, and the approximated transition matrices (**App. H, Fig. S4**)
>
> To conclude, the additional baselines further demonstrate LICCHIE’s advantages. Namely, in contrast to SLDS, which simplifies temporal changes through abrupt switches with a single active dynamics per interval, and GNN, which learns implicit black-box representations, LICCHIE captures smoothly changing interactions in time and space. Since interaction matrices decompose into parsimonious rank-1 components with cell-, distance-, and time-varying weights, multiple interaction mechanisms can be co-present and active simultaneously.
>
> **Revision:** We included two additional baselines methods evaluated on the simulated data:
>
> 1. Per–cell Switching Linear Dynamical Systems (SLDS).
>
> 2. Graph Neural Networks (GNNs)
>
> The revised manuscript includes a detailed one-to-one comparison (**App. H**) quantitative evaluations (**Fig. 2, Fig. S2**) and qualitative visualizations (**Fig S3, Fig. S4**).
>
> 1. Linderman, S., Antin, B., Zoltowski, D., & Glaser, J. (2020). SSM: Bayesian Learning and Inference for State Space Models (Version 0.0.1) [Computer software]  https://github.com/lindermanlab/ssm.
>
> [2] **Real-world data comparisons**
>
> **Concern:** _Comparisons with baselines on the real data are missing._
>
> **Clarification:** Thank you for pointing this out. On real data application our focus was biological interpretation, leading to novel insight based on the inferred interaction motifs; since ground truth interactions are unavailable in the real data, comparisons to baselines are predominantly qualitative in nature or are limited to checking component statistics (e.g., sparsity, localized vs. distributed structure) without knowing the true underlying operators. With that we agree with the reviewer that exploring components identified by other methods can enhance our work, and hence following the inclusion of the two new baselines for the synthetic data (SLDS and GNN; **Fig. S3, Fig S4**), we will incorporate such analysis for the real-world data.
>
> **Revision:** We are working on baseline comparisons over the real-world data.

---

> > ### Author Response · Authors · 2025-11-24
> >
> > **Questions:**
> >
> > [1] **Alternative tensor decomposition approaches**
> >
> > **Concern**: _Does it make biological sense to consider a tensor decomposition with a core tensor instead (Tucker decomposition)?_
> >
> > **Response**: Thank you for this suggestion. The objective of the PARAFAC tensor decomposition step is to infer the underlying vectors that define the rank-1 components. These components can be directly interpreted as source and target effects, and this way, this architecture allows for interpretability, which is a key driving force for the development of LICCHIE.
> >
> > Applying Tucker decomposition instead would yield components that cannot be directly interpreted in terms of the underlying interactions we seek to identify.
> >
> > For example, the core tensor in Tucker decomposition (e.g. the tensor $G \in \mathbb{R}^{P \times Q \times R}$ in the decomposition $Y = G \times_1 A \times_2 B \times_3 C$) lacks clear interpretability since its entries represent cross-modal multi-way dependencies.
> >
> > In contrast, PARAFAC's rank-1 components provide direct meaning of which source features affect which target features. For example, considering the vector sets $\{c_j\}$ and $\{r_j\}$ from our paper, each entry $c_j[m]$ can tell us the effect on feature $m$, while each entry $r_j[m']$ tells the contribution of source feature $m'$ to this effect. When taking the outer product between these vectors, which builds the transition matrices, each entry $A_{m,m'}$ tells the effect of $m$ on $m'$, which is at the core of our interpretability. PARAFAC, in contrast to Tucker, allows identifying these vectors without an uninterpretable core, and hence was chosen in our design over Tucker.
> >
> > **Revision**: We have added a remark following the lines of this response to the discussion of the revised submission, clarifying the benefits of the PARAFAC based rank-1 decomposition (**Section 6**).
> >
> > [2] **Definition of “cell state”**
> >
> > **Concern:** _The paper mentions cell state several times, but this seems largely aspirational, unless I misunderstood the details of the real dataset._
> >
> > **Response:** We appreciate the opportunity to clarify this point–the concept of “cell state” is indeed vague and thus we understand the necessity to properly define it with respect to a given setting. That said, this term is broadly used as a generic characterization of a cell and can range in its granularity and details it captures.
> >
> > In our work, “cell state” denotes the imaging-derived feature vector: a multi-dimensional summary of each cell at time $t$ constructed from live-cell image features (e.g. morphology, intensity, motion, and local spatial context). LICCHIE in turn models how neighbors’ features at time $t-1$ influences a target cell’s imaging-derived state at time $t$. Consequently, we do not claim to infer/refine the cell state representation within our modeling framework.
> >
> > **Concern:** _Can you clarify what “cell state” means in the real data application presented here?_
> >
> > **Response:** For the NK–tumor organoid application [2], the state vector comprises morphological, spatial, and intensity descriptors extracted from the 3D time-lapse volumes, together with population-specific attributes provided by the dataset authors—specifically, an NK polarization readout and a tumor-cell mitotic phase label. Throughout the manuscript, references to “cell state” therefore refer to this observed, imaging-based representation (**Section 5.2, App. F**).
> >
> > **Concern:** _If you’re not modeling it here, what sort of data would you need to do true cell state modeling?_
> >
> > **Response:** We are indeed not modeling cell state here yet the notion of “true” cell state is also not well defined. More comprehensive measurements of molecular information could improve the characterization of cell state but are often destructive, hence providing only a snapshot, thus lacking the temporal resolution of interactions.
> >
> > **Revision:** We included a clarification, defining our usage of cell state within the LICCHIE framework (**Section 3**) and extend the discussion on possible extensions of LICCHIE to incorporate static measurements, providing a more accurate cell state representation, by relying on trajectory inference methods to recover the temporal axis (**Section 6**).
> >
> > 2. Liu et al. Deep Learning–Based 3D Single-Cell Imaging Analysis Pipeline Enables Quantification of Cell–Cell Interaction Dynamics in the Tumor Microenvironment. Cancer Research 84.4: 517-526 (2024).

---

> > ### Author Response · Authors · 2025-12-03
> >
> > Of note, the revised submission now includes **Real-world data comparisons**:
> >
> > **Clarification**:  An in-depth comparative evaluation on the 3D live-cell imaging dataset considering per-cell SLDS, GNNs and LICCHIE assessing biological relevance through prediction and classification accuracies. The analysis demonstrates that LICCHIE’s outputs allow for direct more accurate prediction of biological attributes.
> >
> > **Revision**: A thorough per method outputs (per-cell SLDS, GNNs and LICCHIE; **Section 5.2, App I, Fig. S8, S9, S10**).

---

### Author Response · Authors · 2025-12-03

Dear AC and Reviewers,

We regret the turnout of events that prevented discussion, but would like to sincerely thank the reviewers for the time and effort spent reviewing our manuscript. We were happy to read that all reviewers found the biological problem well formulated and motivated _“biologically grounded and is well-motivated”_, and appreciated the formulation of the framework _“compact and well-structured”_.

We believe we were able to **address the concerns** raised by the reviewers, and are grateful for their feedback and suggestions, which **substantially improved the manuscript**.

**Here is a summary of the points we addressed in the rebuttal:**

* **Extended model evaluations:** Added two baseline methods, *Switching Linear Dynamical Systems (SLDS)* and *Graph Neural Networks (GNNs)*–confirming LICCHIE's superior recovery of ground truth structure (**Reviewer** **J9Kt**).
* **Real-world baselines**: Performed an in-depth evaluation of SLDS, GNNs versus our method,  LICCHIE, on the 3D live-cell imaging dataset, demonstrating LICCHIE’s outputs allow for direct, more accurate, prediction of biological attributes (**Reviewer J9Kt**).
* **Temporal consistency via feature space:** Alleviated the ambiguity in text–clarifying that LICCHIE enforces temporal consistency through *short-term dependencies (t-1 to t)* for the same cell, and works in feature space to aggregate across distant time points (**Reviewer** **j8Lv**).
* **Interpretability design:** Detailed the advantages of PARAFAC decomposition over possible alternatives (e.g., Tucker decomposition), emphasizing that PARAFAC’s rank-1 components provide direct interpretability as source and target effects (**Reviewer** **J9Kt**).
* **Biological validity:** Contextualized interpretations within the 3D tumor organoid/NK system, clarifying that the focus on time-varying, local influences aligns with the dominant mode of brief, contact-dependent cytotoxic action in this setting (**Reviewer** **Siuv**).
* **Validation & mechanism:** Clarified that phenotypic validation is attained by assessing inferred weights on held-out phenotypes. Explained more clearly how LICCIE’s design enables *future work* to infer molecular mechanism mapping (**Reviewer** **Siuv**).
* **Definition of cell state:** Providing a definition of "cell state" within the LICCHIE framework –imaging-derived feature vector used in the model. Relate to possible future extensions to support molecular features (**Reviewer** **J9Kt**).

**Additions to our revision include the following items:**

* **New baseline comparisons:** Included *Per-cell SLDS* and *GNNs* evaluation over:
  * ***Simulated data*** **:** Quantitative and qualitative assessment of *per-cell SLDS* and *GNNs* performance over the synthetic dataset, including detailed per-method comparison to LICCHIE **(Sec. 5.1, App. H, Fig. 3, Fig. S2, Fig. S3, Fig S4**; **Reviewer J9Kt**).
  * ***Real-world application*****:** Thorough evaluation of per method outputs (*per-cell SLDS, GNNs* and *LICCHIE*), assessing biological relevance through prediction and classification accuracies (**Sec. 5.2, App I, Fig. S8, S9, S10; Reviewer J9Kt**),
* **Key definitions:** Added the definition of *Natural Killer cells (NK)* to the abstract (**Abstract**; **Reviewer j8Lv**).
* **Methodological justifications:** Added remarks clarifying the benefits of the *PARAFAC rank-1 decomposition* for interpretability (**Sec. 6**; **Reviewer J9Kt**).
* **Clarity on dynamics:** Clarified the rationale for operating in feature space while maintaining short-range temporal consistency (**Fig. 1, Sec. 1, Sec. 3, Sec. 4, Sec. 6**; **Reviewer** **j8Lv**).
* **Cell state definition:** Included a clarification defining the usage of "cell state" (**Sec. 3**) and extended the discussion on incorporating static molecular measurements (**Sec. 6**; **Reviewer J9Kt**).
* **Biological context:** Added contextualization to the real-data application (**Sec. 5.2**) and emphasized analyzing interpretations relative to the *experimental design* (**Sec. 6**; **Reviewer Siuv**).
* **Future work:** Added language noting that molecular mechanism mapping is *future work* and advocating for multi-omic integration using LICCHIE as a scaffold (**Sec. 6**; **Reviewer Siuv**).

---

### Meta-Review · Area_Chair_9GRv · 2026-01-05

**Summary:**

After the rebuttal, one reviewer still has some major concerns. The paper should be further revised.

**Reviewer Scores:**

n/a

---

### Decision · Program_Chairs · 2026-01-26

Reject